# MINIMAX CURRICULUM LEARNING: MACHINE TEACHING WITH DESIRABLE DIFFICULTIES AND SCHEDULED DIVERSITY

**Tianyi Zhou & Jeff Bilmes**
University of Washington, Seattle
{tianyizh,bilmes}@uw.edu

## ABSTRACT

We introduce and study minimax curriculum learning (MCL), a new method for adaptively selecting a sequence of training subsets for a succession of stages in machine learning. The subsets are encouraged to be small and diverse early on, and then larger, harder, and allowably more homogeneous in later stages. At each stage, model weights and training sets are chosen by solving a joint continuous-discrete minimax optimization, whose objective is composed of a continuous loss (reflecting training set hardness) and a discrete submodular promoter of diversity for the chosen subset. MCL repeatedly solves a sequence of such optimizations with a schedule of increasing training set size and decreasing pressure on diversity encouragement. We reduce MCL to the minimization of a surrogate function handled by submodular maximization and continuous gradient methods. We show that MCL achieves better performance and, with a clustering trick, uses fewer labeled samples for both shallow and deep models. Our method involves repeatedly solving constrained submodular maximization of an only slowly varying function on the same ground set. Therefore, we develop a heuristic method that utilizes the previous submodular maximization solution as a warm start for the current submodular maximization process to reduce computation while still yielding a guarantee.

## 1 INTRODUCTION

Inspired by the human interaction between teacher and student, recent studies (Khan et al., 2011; Basu & Christensen, 2013; Spitkovsky et al., 2009) support that learning algorithms can be improved by updating a model on a designed sequence of training sets, i.e., a curriculum. This problem is addressed in curriculum learning (CL) (Bengio et al., 2009), where the sequence is designed by a human expert or heuristic before training begins. Instead of relying on a teacher to provide the curriculum, self-paced learning (SPL) (Kumar et al., 2010; Tang et al., 2012a; Supancic III & Ramanan, 2013; Tang et al., 2012b) chooses the curriculum during the training process. It does so by letting the student (i.e., the algorithm) determine which samples to learn from based on their hardness. Given a training set $\mathcal{D} = \{(x_1, y_1), \ldots, (x_n, y_n)\}$ of $n$ samples and loss function $L(y_i, f(x_i, w))$, where $x_i \in \mathbb{R}^m$ represents the feature vector for the $i^{th}$ sample, $y_i$ is its label, and $f(x_i, w)$ is the predicted label provided by a model with weight $w$, SPL performs the following:

$$\min_{w \in \mathbb{R}^m} \min_{\nu \in [0,1]^n} \left[ \sum_{i=1}^{n} \nu_i L\left(y_i, f(x_i, w)\right) - \lambda \sum_{i=1}^{n} \nu_i \right]. \tag{1}$$

SPL jointly learns the model weights $w$ and sample weights $\nu$, which end up being 0-1 indicators of selected samples, and it does so via alternating minimization. Fixing $w$, minimization w.r.t. $\nu$ selects samples with loss $L(y_i, f(x_i, w)) < \lambda$, where $\lambda$ is a "hardness parameter" as it corresponds to the hardness as measure by the current loss (since with large $\lambda$, samples with greater loss are allowed in). Self-paced curriculum learning (Jiang et al., 2015) introduces a blending of "teacher mode" in CL and "student mode" in SPL, where the teacher can define a region of $\nu$ by attaching a linear constraint $a^T \nu \leq c$ to Eq. (1). SPL with diversity (SPLD) (Jiang et al., 2014), adds to Eq. (1) a negative group sparse regularization term $-\gamma \|\nu\|_{2,1} \triangleq -\gamma \sum_{j=1}^{b} \|\nu^{(j)}\|_2$, where the samples are

divided into $b$ groups beforehand and $\nu^{(j)}$ is the weight vector for the $j^{th}$ group. Samples coming from different groups are thus preferred, to the extent that $\gamma > 0$ is large.

CL, SPL, and SPLD can be seen as a form of continuation scheme (Allgower & Georg, 2003) that handles a hard task by solving a sequence of tasks moving from easy to hard; the solution to each task is the warm start for the next slightly harder task. That is, each task, in the present case, is determined by the training data subset and other training hyperparameters, and the resulting parameters at the end of a training round are used as the initial parameters for the next training round. Such continuation schemes can reduce the impact of local minima within neural networks (Bengio et al., 2013; Bengio, 2014). With SPL, after each round of alternating minimization to optimize Eq. (1), $\lambda$ is increased so that the next round selects samples that have a larger loss, a process (Khan et al., 2011; Tang et al., 2012b; Basu & Christensen, 2013) that can both help avoid local minima and reduce generalization error. In SPLD, $\gamma$ is also increased between training rounds, increasingly preferring diversity. In each case, each round results in a fully trained model for the currently selected training samples.

Selection of training samples has been studied in other settings as well, often with a different motivation. In active learning (AL) (Settles, 2010) and experimental design (Montgomery, 2006), the learner can actively query labels of samples from an unlabeled pool during the training process, and the goal is to reduce annotation costs. The aim is to achieve the same or better performance using fewer labeled samples by ruling out uninformative ones. Diversity modeling was introduced to AL in (Wei et al., 2015). It uses submodular maximization to select diverse training batches from the most uncertain samples. However, changing the diversity during the learning process has not been investigated as far as we know. In boosting (Schapire, 1990; Freund & Schapire, 1997), the goal is to learn an ensemble of weak classifiers sequentially; it does this by assigning weights to all samples, with larger weights given to samples having larger loss measured by an aggregation of previously trained models. Both active learning and boosting favor samples that are difficult to predict, since they are the most informative to learn. For example, uncertainty sampling (Culotta & McCallum, 2005; Scheffer et al., 2001; Dagan & Engelson, 1995; Dasgupta & Hsu, 2008) selects samples that are most uncertain, while query by committee (Seung et al., 1992; Dagan & Engelson, 1995; Abe & Mamitsuka, 1998) selects the ones that multiple models most disagree on. With machine teaching (Khan et al., 2011; Zhu, 2015; Patil et al., 2014; Zhu et al., 2018), a separate teacher helps the training procedure find a good model.

The SPL approach starts with a smaller set of easy samples and gradually increases the difficulty of the chosen samples as measured by the sample loss of the model produced by previous round's training. One of the difficulties of this approach is the following: since for any given value of $\lambda$ the relatively easiest samples are chosen, there is a good chance that the process can repeatedly select a similar training set over multiple rounds and therefore can learn slowly. This is precisely the problem that SPLD address — by concomitantly increasing the desired diversity over rounds, the sample selection procedure chooses from an increasingly diverse set of different groups, as measured by $\|\nu\|_{2,1}$. Therefore, in SPLD, early stages train on easier not necessarily diverse samples and later stages train on harder more diverse samples.

There are several challenges remaining with SPLD, however. One is that in early stages, it is still possible to repeatedly select a similar training set over multiple rounds since diversity might not increase dramatically between successive rounds. Potentially more problematically, it is not clear that having a large diversity selection weight in late stages is desirable. For example, with a reasonably trained model, it might be best to select primarily the hardest samples in the part of the space near the difficult regions of the decision boundaries. With a high diversity weight, samples in these difficult decision boundary regions might be avoided in favor of other samples perhaps already well learnt and having a large margin only because they are diverse, thereby leading to wasted effort. At such point, it would be beneficial to choose points having small margin from the same region but that might not have the greatest diversity, especially when using only a simple notion of diversity such as the group sparse norm $\|v\|_{2,1}$. Also, it is possible that late stages of learning can select outliers only because they are both hard and diverse. Lastly, the SPL/SPLD min-min optimization involves minimizing a lower bound of the loss, while normally one would, if anything, wish to minimize the loss directly or at least an upper bound.

Motivated by these issues, we introduce a new form of CL that chooses the hardest diverse samples in early rounds of training and then actually *decreases*, rather than increases, diversity as training rounds proceed. Our contention is that diversity is more important during the early phases of training when

only relatively few samples are selected. Later rounds of training will naturally have more diversity opportunity simply because the size of the selected samples is much larger. Also, to avoid successive rounds selecting similar sets of samples, our approach selects the hardest, rather than the easiest, samples at each round. Hence, if a set of samples is learnt well during one training round, those samples will tend to be ill-favored in the next round because they become easier. We also measure hardness via the loss function, but the selection is always based on the hardest and most diverse samples of a given size $k$, where the degree of diversity is controlled by a parameter $\lambda$, and where diversity is measured by an arbitrary non-monotone submodular function. In fact, for binary variables the group sparse norm is also submodular where $\|\nu\|_{2,1} = \sum_{j=1}^{b} \sqrt{|C_j \cap A|} = F(A)$ where $A$ is the set for which $\nu$ is the characteristic vector, and $C_j$ is the set of samples in the $j^{\text{th}}$ group. Our approach allows the full expressive class of submodular functions to be used to measure diversity since the selection phases is based on submodular optimization.

Evidence for the naturalness of such hardness and diversity adjustment in a curriculum can also be found in human education. For example, courses in primary school usually cover a broad, small, and relatively easy range of topics, in order to expose the young learner to a diversity of knowledge early on. In college and graduate school, by contrast, students focus on advanced deeper knowledge within their majors. As another example, studies of bilingualism (Bialystok et al., 2012; Li et al., 2014; Mechelli et al., 2004; Kovács & Mehler, 2009) show that learning multiple languages in childhood is beneficial for future brain development, but early-age multi-lingual learning is usually not advanced or concentrated linguistically for any of the languages involved. Still other studies argue that difficulty can be desired at early human learning stages (Bjork & Bjork, 1992; McDaniel & Butler, 2011).

### 1.1 Our Approach: Minimax Curriculum Learning

We introduce a new form of curriculum learning called minimax curriculum learning (MCL). MCL increases desired hardness and reduces diversity encouragement over rounds of training. This is accomplished by solving a sequence of minimax optimizations, each of which having the form:

$$\min_{w \in \mathbb{R}^m} \max_{A \subseteq V, |A| \leq k} \sum_{i \in A} L\left(y_i, f(x_i, w)\right) + \lambda F(A). \tag{2}$$

The objective is composed of the loss on a subset $A$ of samples evaluating their hardness and a normalized monotone non-decreasing submodular function $F : 2^V \to \mathbb{R}_+$ measuring $A$'s diversity, where $V$ is the ground set of all available samples. A larger loss implies that the subset $A$ has been found harder to learn, while a larger $F(A)$ indicates greater diversity. The weight $\lambda$ controls the trade-off between hardness and diversity, while $k$, the size of the resulting $A$, determines the number of samples to simultaneously learn and hence is a hardness parameter.

It is important to realize that $F(A)$ is not a parameter regularizer (e.g., $\ell_1$ or $\ell_2$ regularization on the parameters $w$) but rather an expression of preference for a diversity of training samples. In practice, one would add to Eq. (2) an appropriate parameter regularizer as we do in our experiments (Section 3).

Like SPL/SPLD, learning rounds are scheduled, here each round with increasing $k$ and decreasing $\lambda$. Unlike SPL/SPLD, we explicitly schedule the number of selected samples via $k$ rather than indirectly via a hardness parameter. This makes sense since we are always choosing the hardest $k$ samples at a given $\lambda$ diversity preference, so there is no need for an explicit real-valued hardness parameter as in SPL/SPLD. Also, the MCL optimization minimizes an upper bound of the loss on any size $k$ subset of training samples.

The function $F(\cdot)$ may be chosen from the large expressive family of submodular functions, all of which are natural for measuring diversity, and all having the following diminishing returns property: given a finite ground set $V$, and any $A \subseteq B \subseteq V$ and a $v \notin B$,

$$F(v \cup A) - F(A) \geq F(v \cup B) - F(B). \tag{3}$$

This implies $v$ is no less valuable to the smaller set $A$ than to the larger set $B$. The marginal gain of $v$ conditioned on $A$ is denoted $f(v|A) \triangleq f(v \cup A) - f(A)$ and reflects the importance of $v$ to $A$. Submodular functions (Fujishige, 2005) have been widely used for diversity models (Lin et al., 2009; Lin & Bilmes, 2011; Batra et al., 2012; Prasad et al., 2014; Gillenwater et al., 2012; Iyer & Bilmes, 2015; Bilmes & Bai, 2017).

Although Eq. (2) is a hybrid optimization involving both continuous variables $w$ and discrete variables $A$, it can be reduced to the minimization of a piecewise function, where each piece is defined by a subset $A$ achieving the maximum in a region around $w$. Each piece is convex when the loss is convex, so various off-the-shelf algorithms can be applied once $A$ has been computed. However, the number of possible sets $A$ is $\binom{n}{k}$, and enumerating them all to find the maximum is intractable. Thanks to submodularity, fast approximate algorithms (Nemhauser et al., 1978; Minoux, 1978; Mirzasoleiman et al., 2015) exist to find an approximately optimal $A$. Therefore, the outer optimization over $w$ will need to minimize an approximation of the piecewise function defined by an approximate $A$ computed via submodular maximization.

## 2 Minimax Curriculum Learning and Machine Teaching

The minimax problem in Eq. (2) can be seen as a two-person zero-sum game between a teacher (the maximizer) and a student (the minimizer): the teacher chooses training set $A$ based on the student's feedback about the hardness (i.e., the loss achieved by current model $w$) and how diverse according to the teacher ($\lambda F(A)$), while the student updates $w$ to reduce the loss on training set $A$ (i.e., learn $A$) given by the teacher. Similar teacher-student interaction also exist in real life. In addition, the teacher usually introduces concepts at the beginning and asks a small number of easy questions from a diverse range of topics and receives feedback from the student, and then further trains the student on the topics the student finds difficult while eschewing topics the student has mastered.

MCL's minimax formulation is different from the min-min formulation used in SPL/SPLD. For certain losses and models, $L(y_i, f(x_i, w))$ is convex in $w$. The min-min formulation, however, is only bi-convex and requires procedures such as alternative convex search (ACS) as in (Bazaraa et al., 1993). Furthermore, diversity regularization of $\nu$ in SPLD leads to the loss of bi-convexity altogether.

Minimizing the worst case loss, as in MCL, is a widely used strategy in machine learning (Lanckriet et al., 2003; Farnia & Tse, 2016; Shalev-Shwartz & Wexler, 2016) to achieve better generalization performance and model robustness, especially when strong assumptions cannot be made about the data distribution. Compared to SPL/SPLD, MCL is also better in that the outer minimization over $w$ in Eq. (2) is a convex program, and corresponds to minimizing the objective $g(w)$ in Eq. (4). On the other hand, querying $g(w)$ requires submodular maximization which can only be solved approximately.

The goal of this section, therefore, is to address the minimax problem in Eq. (2), i.e., the minimization $\min_{w \in \mathbb{R}^m} g(w)$ of the following objective $g(w)$.

$$g(w) \triangleq \max_{A \subseteq V, |A| \leq k} \sum_{i \in A} L\left(y_i, f(x_i, w)\right) + \lambda F(A) \tag{4}$$

If the loss function $L(y_i, f(x_i, w))$ is convex w.r.t. $w$, then $g(w)$ is convex but, as mentioned above, enumerating all subsets is intractable. Defining the discrete objective $G_w : 2^V \to \mathbb{R}_+$ where

$$G_w(A) \triangleq \sum_{i \in A} L\left(y_i, f(x_i, w)\right) + \lambda F(A). \tag{5}$$

shows that computing $g(w)$ in involves a discrete optimization over $G_w(A)$, a problem that is submodular since $G_w(A)$ is weighted sum of a non-negative (since loss is non-negative) modular and a submodular function, and thus $G_w$ is monotone non-decreasing submodular. Thus, the fast greedy procedure mentioned earlier can be used to approximately optimizes $G_w(A)$ for any $w$.

Let $\hat{A}_w \subseteq V$ be the $k$-constrained greedy approximation to maximizing $G_w(A)$. We define the following approximate objective:

$$\hat{g}(w) \triangleq \sum_{i \in \hat{A}_w} L\left(y_i, f(x_i, w)\right) + \lambda F(\hat{A}), \tag{6}$$

and note that it satisfies $\alpha g(w) \leq \hat{g}(w) \leq g(w)$ where $\alpha$ is the approximation factor of submodular optimization. For $\tilde{w}$ within a region around $w$, $\hat{g}(\tilde{w})$ will utilize the same set $\hat{A}_w$. Therefore, $\hat{g}(w)$ is piecewise convex, if the loss function $L(y_i, f(x_i, w))$ is convex w.r.t. $w$, and different regions of within $\mathbb{R}^m$ are associated with different $\hat{A}$ although not necessarily the same regions or sets that define $g(w)$. We show in Section 2.2 that minimizing $\hat{g}(w)$ offers an approximate solution to Eq. (2).

With $\hat{g}(w)$ given, our algorithm is simply gradient descent for minimizing $\hat{g}(w)$, where many off-the-shelf methods can be invoked, e.g., SGD, momentum methods, Nesterov's accelerated gradient (Nesterov, 2005), Adagrad (Duchi et al., 2011), etc. The key problem is how to obtain $\hat{g}(w)$, which depends on suboptimal solutions in different regions of $w$. It is not necessary, however, to run submodular maximization for every region of $w$. Since we use gradient descent, we only need to know $\hat{g}(w)$ for $w$ on the optimization path. At the beginning of each iteration, we fix $w$ and use submodular maximization to achieve the $\hat{A}_w$ that defines $\hat{g}(w)$. Then a gradient update step is applied to $\hat{g}(w)$. Let $A_w^*$ represent the optimal solution to Eq. (5), then $\hat{A}_w$ satisfies $G(\hat{A}) \geq \alpha G(A^*)$.

---

**Algorithm 1** Minimax Curriculum Learning (MCL)

1: **input:** $\pi(\cdot, \eta), \gamma, p, \Delta, \tilde{\alpha}$
2: **output:** $w_T^0$
3: **initialize:** $\tau \leftarrow 1, w_\tau^0, \lambda, k,$
4: **while** *not "converged"* **do**
5:   **for** $t \in \{0, \cdots, p\}$ **do**
6:     $G(A) \leftarrow \sum_{i \in A} L\left(y_i, f(x_i, w_\tau^t)\right) + \lambda F(A)$;
7:     $\hat{A} \leftarrow$ WS-SUBMODULARMAX$(G, k, \hat{A}, \tilde{\alpha})$;
8:     $\nabla \hat{g}(w_\tau^t) = \frac{\partial}{\partial w} \sum_{i \in \hat{A}} L\left(y_i, f(x_i, w_\tau^t)\right)$;
9:     $w_\tau^{t+1} \leftarrow w_\tau^t + \pi\left(\{w_\tau^{1:t}\}, \{\nabla \hat{g}(w_\tau^{1:t})\}, \eta\right)$;
10:   **end for**
11:   $w_{\tau+1}^0 \leftarrow w_\tau^p, \lambda \leftarrow (1-\gamma) \cdot \lambda, k \leftarrow k+\Delta, \tau \leftarrow \tau+1$;
12: **end while**

---

Algorithm 1 details MCL. Lines 5-10 solve the optimization in Eq. (2) with $\lambda$ and $k$ scheduled in line 11. Lines 6-7 finds an approximate $\hat{A}$ via submodular maximization, discussed further in Section 2.1. Lines 8-9 update $w$ for the current $\hat{A}$ by gradient descent $\pi(\cdot, \eta)$ with learning rate $\eta$. The inner optimization stops after $p$ steps and then $\lambda$ is reduced by factor $1 - \gamma$ where $\gamma \in [0, 1]$ and $k$ is increased by $\Delta$. The outer optimization stops after $T$ steps when a form of "convergence", described below, is achieved. Given $\hat{A}_w$, $\hat{g}(w)$ has gradient

$$\nabla \hat{g}(w) = \frac{\partial}{\partial w} \sum_{i \in \hat{A}_w} L\left(y_i, f(x_i, w)\right), \quad (7)$$

and thus gradient descent method can update $w$. For example, we can treat $\hat{A}$ as a batch if $k$ is small, and update $w$ by $w \leftarrow w - \eta \nabla \hat{g}(w)$ with learning rate $\eta$. For large $\hat{A}_w$, we can use SGD that applies an update rule to mini-batches within $\hat{A}_w$. More complex gradient descent rules $\pi(\cdot, \eta)$ can take historical gradients and $w_\tau^t$'s into account leading to $w^{t+1} \leftarrow w^t + \pi\left(\{w^{1:t}\}, \{\nabla \hat{g}(w^{1:t})\}, \eta\right)$.

Considering the outer loop as well, the algorithm approximately solves a sequence of Eq. (2)s with decreasing $\lambda$ and increasing $k$, where the previous solutions act as a warm start for the next iterations. This corresponds to repeatedly updating the model $w$ on a sequence of training sets $\hat{A}$ that changes from small, diverse, and hard to large.

## 2.1 SUBMODULAR MAXIMIZATION

Although solving Eq. (5) exactly is NP-hard, a near-optimal solution can be achieved by the greedy algorithm, which offers a worst-case approximation factor of $\alpha = 1 - e^{-1}$ (Nemhauser et al., 1978). The algorithm starts with $A \leftarrow \emptyset$, and selects next the element with the largest marginal gain $f(v|A)$ from $V \backslash A$, i.e., $A \leftarrow A \cup \{v^*\}$ where $v^* \in \arg\max_{v \in V \backslash A} f(v|A)$, and this repeats until $|A| = k$. It is simple to implement, fast, and usually outperforms other methods, e.g., those based on integer linear programming. It requires $\mathcal{O}(nk)$ function evaluations for ground set size $|V| = n$. Since Algorithm 1 runs greedy $Tp$ times, it is useful for the greedy procedure to be as fast as possible. The accelerated, or lazy, greedy algorithm (Minoux, 1978) reduces the number of evaluations per step by updating a priority queue of marginal gains, while having the same output and guarantee as the original (thanks to submodularity) and offers significant speedups. Still faster variants are also available Mirzasoleiman et al. (2015; 2016). Our own implementation takes advantage of the fact that line 7 of Algorithm 1 repeatedly solves submodular maximization over a sequence of submodular functions that are changing only slowly, and hence the previous set solution can be used as a warm start for the current algorithm, a process we call WS-SUBMODULARMAX outlined in Algorithm 2.

The greedy procedure offers much better approximation factors than $1 - e^{-1}$ when the objective $G(A)$ is close to modular. Specifically, the approximation factor becomes $\alpha = (1 - e^{-\kappa_G})/\kappa_G$ (Conforti & Cornuejols, 1984), which depends on the curvature $\kappa_G \in [0, 1]$ of $G(A)$ defined as

$$\kappa_G \triangleq 1 - \min_{j \in V} \frac{G(j|V \backslash j)}{G(j)}. \quad (8)$$

When $\kappa_G = 0$, $G$ is modular, and when $\kappa_G = 1$, $G$ is fully curved and the above bound recovers $1 - e^{-1}$. $G(A)$ becomes more modular as the outer loop proceeds since $\lambda$ decreases. Therefore, the approximation improves with the number of outer loops. In fact, we have:

**Lemma 1.** *Let $G(A) = L(A) + \lambda F(A)$ where $F$ is a monotone non-decreasing submodular function with curvature $\kappa_F$, $L$ is a non-negative modular function, and $\lambda \geq 0$. Then $\kappa_G \leq \kappa_F / (c_1/\lambda + 1)$ where $c_1 = \min_{j \in V} L(j)/F(j)$.*

The proof is given in Appendix 4.1. In MCL, therefore, the submodular approximation improves ($\alpha \to 1$) as $\lambda$ grows, and the surrogate function $\hat{g}(w)$ correspondingly approaches the true convex objective $g(w)$.

## 2.2 Conditions at Convergence

In this section, we study how close the solution $\hat{w}$ is of applying gradient descent to $\hat{g}(w)$, where we assume $p$ is large enough so that a form of convergence occurs. Specifically, in Theorem 1, we analyze the upper bound on $\|\hat{w} - w^*\|_2^2$ based on two assumptions: 1) the loss $L(y_i, f(x_i, w))$ being $\beta$-strongly convex w.r.t. $w$; and 2) $\hat{w}$ is achieved by running gradient descent in lines 6-9 of Algorithm 1 until convergence, defined as the gradient reaching zero. In case the loss $L(y_i, f(x_i, w))$ is convex but not $\beta$-strongly convex, a commonly used trick to modify it to $\beta$-strongly convex is to add an $\ell_2$ regularization $(\beta/2)\|w\|_2^2$. In addition, for non-convex $L(y_i, f(x_i, w))$, it is possible to prove that with high probability, a noise perturbed SGD on $\hat{g}(w)$ can hit an $\epsilon$-optimal local solution of $g(w)$ in polynomial time — we leave this for future work. In our empirical study (Section 3), MCL achieves good performance even when applied to non-convex deep neural networks. The following theorem relies on the fact that the maximum of multiple $\beta$-strongly convex functions is also $\beta$-strongly convex, shown in Appendix 4.2.

**Theorem 1** (Inner-loop convergence)**.** *For the minimax problem in Eq. (2) with ground set of samples $V$ and $\lambda \geq 0$, if the loss function $L(y_i, f(x_i, w))$ is $\beta$-strongly convex and $|V| \geq k$, running lines 6-9 of Algorithm 1 until convergence (defined as the gradient reaching zero) yields a solution $\hat{w}$ satisfying*

$$\|\hat{w} - w^*\|_2^2 \leq \frac{2}{k\beta} \left( \frac{1}{\alpha} - 1 \right) \cdot g(w^*), \tag{9}$$

*$\hat{w}$ is the solution achieved at convergence, $w^*$ is the optimal solution of the minimax problem in Eq.(2), $g(w^*)$ is the objective value achieved on $w^*$, and $\alpha$ is the approximation factor that submodular maximization can guarantee for $G(A)$.*

The proof is given in Appendix 4.3.

It is interesting to note that the bound depends both on the strong convexity parameter $\beta$ and on the submodular maximization approximation $\alpha$. As mentioned in Lemma 1, as $\lambda$ gets smaller, the approximation factor $\alpha$ approaches 1 meaning that the bound in Equation (9) improves.

We mention the convergence criteria where the gradient reaches zero. While it is possible, in theory, for lines 6-9 of Algorithm 1 to oscillate amongst the non-differentiable boundaries between the convex pieces, with most damped learning rates, this will eventually subside and the algorithm will remain within one convex piece. The reason for this is line 7 of the algorithm always chooses one $\hat{A}$ thereby selecting one convex piece associated with the region around $w_\tau^t$, and with only small subsequent adjustments to $w_\tau^t$, the same $\hat{A}$ will continue to be selected. Hence, the algorithm will, in such case, reach the minimum of that convex piece where the gradient is zero.

We can restate and then simplify the above bound in terms of the resulting parameters, and corresponding $\lambda, k$ values, used at a particular iteration $\tau$ of the outer loop. In the following, $\hat{w}_\tau$ is the solution achieved by Algorithm 1 at the iteration $\tau$ of the outer loop, and the optimal solution of the minimax problem in Eq.(2) with $\lambda, k$ set as in iteration $\tau$ is denoted $w_T^*$.

**Corollary 1.** *If the loss function $L(y_i, f(x_i, w))$ is $\beta$-strongly convex, the submodular function $F(\cdot)$ has curvature $\kappa_F$, and if each inner-loop in Algorithm 1 runs until convergence, then the solution $\hat{w}_\tau$ at the end of the $\tau^{th}$ iteration of the outer-loop fulfills:*

$$\|\hat{w}_\tau - w_\tau^*\|_2^2 \leq \frac{2\kappa_F}{k\beta(c_1/\lambda + 1)} g(w_\tau^*) \leq \frac{2\kappa_F}{\beta c_1} \times \frac{\lambda}{k} \times g(w_\tau^*), \tag{10}$$

*where $w_\tau^*$ is the optimal solution of the minimax problem in Eq.* (2) *with $\lambda$ set as in the $\tau^{th}$ outer loop iteration.*

*Thus, if $k$ starts from $k_0$ and linearly increases via $k \leftarrow k + \Delta$ (as in line 11 of Algorithm alg:mcl),*

$$\|\hat{w}_\tau - w_\tau^*\|_2^2 \leq \frac{2\kappa_F \lambda_0}{\beta c_1} \times \frac{(1-\gamma)^\tau}{(k_0 + \tau\Delta)} \times [g(w_\infty^*) + \lambda_0 c_2 (1-\gamma)^\tau], \tag{11}$$

*Otherwise, if $k$ increases exponentially, i.e., $k \leftarrow (1+\Delta) \cdot k$,*

$$\|\hat{w}_\tau - w_\tau^*\|_2^2 \leq \frac{2\kappa_F \lambda_0}{\beta c_1 k_0} \times \left(\frac{1-\gamma}{1+\Delta}\right)^\tau \times [g(w_\infty^*) + \lambda_0 c_2 (1-\gamma)^\tau]. \tag{12}$$

*In the above, $\lambda_0$ and $k_0$ are the initial values for $\lambda$ and $k$, $c_1 = \min_{j \in V, t \in [1,\tau]}[L(y_i, f(x_i, \hat{w}_\tau^t))/F(j)]$, $c_2 = \max_{A \subseteq V, |A| \leq k} F(A)$, and $g(w_\infty^*) = \min_{w \in \mathbb{R}^m} \max_{A \subseteq V, |A| \leq k} \sum_{i \in A} L(y_i, f(x_i, w))$.*

The proof can be found in Appendix 4.5. On the one hand, the upper bound above is in terms of the ratio $\lambda/k$ which improves with larger subset sizes. On the other hand, submodular maximization becomes more expensive with $k$. Hence, Algorithm 1 chooses a schedule to decrease $\lambda$ exponentially and increase $k$ only linearly. Also, we see that the bound is dependent on the submodular curvature $\kappa_F$, the strongly-convex constant $\beta$, and $c_1$ which relates the submodular and modular terms (similar to as in Lemma 1). These quantities ($\kappa_F/\beta$ and $c_1$) might be relevant for other convex-submodular optimization schemes.

## 2.3 HEURISTIC IMPROVEMENTS

There are several heuristic improvements we employ that are described next.

Algorithm 1 stops gradient descent after $p$ steps. A reason for doing this is that $\hat{w}^p$ can be sufficient as a warm-start for the next iteration if $p$ is large enough. We also have not observed any benefit for larger $p$, although we do eventually observe convergence empirically when the average loss no longer change appreciably between stages.

Also, lines 6-7 of Algorithm 1 require computing the loss on all the samples, and each step of the greedy algorithm needs to, in the worst case, evaluate the marginal gains of all of the unselected samples. Moreover, this is done repeatedly in the inner-most block of two nested loops. Therefore, we use two heuristic tricks to improve efficiency.

Fist, rather than selecting individual samples, we first cluster the data and then select clusters, thereby reducing the ground set size from the number of samples to the number of clusters. We replace the per-sample loss $L(y_i, f(x_i, w))$ with a per-cluster loss $L(Y^{(i)}, f(X^{(i)}, w))$ that we approximate by the loss of the sample closest to the centroid within each cluster:

$$L\left(Y^{(i)}, f(X^{(i)}, w)\right) \triangleq \sum_{j \in C^{(i)}} L(y_j, f(x_j, w)) \approx |C^{(i)}| L\left(y^{(i)}, f(x^{(i)}, w)\right), \tag{13}$$

where $C^{(i)}$ is the set of indices of the samples in the $i^{\text{th}}$ cluster, and $x^{(i)}$ with label $y^{(i)}$ is the sample closest to the cluster centroid. We find that the loss on $x^{(i)}$ is sufficiently representative to approximately indicate the hardness of the cluster. The set $V$ becomes the set of clusters and $A \subseteq V$ is a set of clusters, and hence the ground set size is reduced speeding up the greedy algorithm. When computing $F(A)$, the diversity of selected clusters, cluster centroids again represent the cluster. In line 8, the gradient is computed on all the samples in the selected clusters rather than on only $x^{(i)}$ at which point the labels of all the samples in the selected clusters are used. Otherwise, when selecting clusters via submodular maximization, the labels of only the centroid samples are needed. Thus, we need only annotate and compute the loss for samples in the selected clusters and the representative centroid samples $x^{(i)}$ of other clusters. This also reduces the need to label all samples up front as only the labels of the selected clusters, and centroid samples of each cluster, are used (i.e., the clustering process itself does not use the labels).

We can further reduce the ground set to save computation during submodular maximization via pre-filtering methods that lead either to no (Wei et al., 2014a) or little (Zhou et al., 2017; Mirzasoleiman et al., 2015) reduction in approximation quality. Moreover, as $\lambda$ decreases in the MCL objective and $G(A)$ becomes more modular, pruning method become more effective. More details are given in Section 4.6.

## 3 EXPERIMENTS

| Method \ Dataset | News20 | MNIST | CIFAR10 | STL10 | SVHN | Fashion |
|---|---|---|---|---|---|---|
| SGD(random) | 14.27 | 0.88 | 18.52 | 21.76 | 5.20 | 7.79 |
| SPL | 15.43 | 1.25 | 21.14 | 20.63 | 5.67 | 7.46 |
| SPLD | 16.23 | 1.18 | 20.79 | 21.25 | 5.40 | 7.80 |
| MCL($\Delta = 0, \lambda = 0, \gamma = 0$) | 15.99 | 1.23 | 18.04 | 20.50 | 5.37 | 7.95 |
| MCL($\Delta = 0, \lambda > 0, \gamma > 0$) | 16.54 | 0.95 | 17.33 | 19.70 | 4.95 | 7.29 |
| MCL($\Delta > 0, \lambda > 0, \gamma = 0$) | 15.45 | 0.82 | 16.93 | 20.40 | 5.29 | 7.07 |
| MCL-RAND | 16.23 | 0.80 | 17.12 | 20.42 | 5.18 | 6.92 |
| MCL($\Delta > 0, \lambda > 0, \gamma > 0$) | **14.12** | **0.75** | **12.87** | **17.83** | **4.19** | **6.36** |

Table 1: Test error (%) for different methods (SGD shows the lowest error out of 10 random trials).

In this section, we apply different curriculum learning methods to train logistic regression models on 20newsgroups (Lang, 1995), LeNet5 models on MNIST (Lecun et al., 1998), convolutional neural nets (CNNs) with three convolutional layers[1] on CIFAR10 (Krizhevsky & Hinton, 2009), CNNs with two convolutional layers [2] on Fashion-MNIST ("Fashion" in all tables) (Xiao et al., 2017), CNNs with six convolutional layers on STL10 (Coates et al., 2011), and CNNs with seven convolutional layers on SVHN (Netzer et al., 2011)[3]. Details on the datasets can be found in Table 3 of the appendix. In all cases, we also use $\ell_2$ parameter regularization on $w$ with weight $1e-4$ (i.e., the weight decay factor of the optimizer).

We compare MCL and its variants to SPL (Kumar et al., 2010), SPLD (Jiang et al., 2014) and SGD with a random curriculum (i.e., with random batches). Each method uses mini-batch SGD for $\pi(\cdot, \eta)$ with the same learning rate strategy to update $w$. The methods, therefore, differ only in the curriculum (i.e., the sequence of training sets).

For SGD, in each iteration, we randomly select 4000 samples (20newsgroups) or 5000 samples (other datasets) and apply mini-batch SGD to the selected samples. In SPL and SPLD, the training set starts from a fixed size $k$ (4000 samples for 20newsgroups, 5000 samples for other datasets), and increases by a factor of $1 + \mu$ (where $\mu = 0.1$) per round of alternating minimization (i.e., per iteration of the outer loop) [4]. We use $\rho$ to denote the number of iterations of the inner loop, which aims to minimize the loss w.r.t. the model $w$ on the selected training set. In SPLD, we also have a weight for the negative group sparsity: it starts from $\xi$ and increases by a factor of 1.1 at each round of alternating minimization (i.e., per iteration of the outer loop). We test five different combinations of $\{\rho, \mu\}$ and $\{\rho, \xi\}$ for SPL and SPLD respectively. The best combination with the smallest test error rate is what we report. Neither SPL nor SPLD uses the clustering trick we applied to MCL: they compute the exact loss on each sample in each iteration. Hence, they have more accurate estimation of the hardness on each sample, and require knowing the labels of all samples (selected and unselected) and cannot reduce annotation costs. Note SPLD still needs to run clustering and use the resulted clusters as groups in the group sparsity (which measures diversity in SPLD). We did not select samples with SPL/SPLD as we do with MCL since we wanted to test SPL/SPLD as originally presented — intuitively, SPL/SPLD should if anything only do better without such clustering due to the more accurate sample-specific hardness estimation. The actual clustering, however, used for SPLD's diversity term is the same as that used for MCL's cluster samples. We apply the mini-batch k-means algorithm to the features detailed in the next paragraph to get the clusters used in MCL and SPLD. Although both SPL and SPLD can be reduced to SGD when $\lambda \to \infty$ (i.e., all samples always selected), we do not include this special case because SGD is already a baseline. For SGD with a random curriculum, results of 10 independent trials are reported.

In our MCL experiments, we use a simple "feature based" submodular function (Wei et al., 2014b) where $F(A) = \sum_{u \in \mathcal{U}} \omega_u \sqrt{c_u(A)}$ and where $\mathcal{U}$ is a set of features. For a subset $A$ of clusters,

---

[1]The "v3" network from `https://github.com/jseppanen/cifar_lasagne`.

[2]A variant of LeNet5 with 64 kernels for each convolutional layer.

[3]The network structures for STL10 and SVHN can be found at `https://github.com/aaron-xichen/pytorch-playground`

[4]Similar to (Jiang et al., 2014), instead of specifying the absolute value of $\lambda$ in each iteration, we find that specifying the number of selected samples $k$ is more robust empirically. Because directly setting $\lambda$ can result in selecting too many or too few samples, but SPL/SPLD needs to increase training samples gradually.

| Dataset | News20 | MNIST | CIFAR10 | STL10 | SVHN | Fashion |
|---|---|---|---|---|---|---|
| Total time | 2649.19s | 3418.97s | 3677.73s | 2953.47s | 34153.81s | 2927.18s |
| WS-SUBMODULARMAX | 62.44s | 35.33s | 127.36s | 206.70s | 1892.62s | 167.55s |

Table 2: Total time (secs.) of MCL($\Delta > 0, \lambda > 0, \gamma > 0$) and time only on WS-SUBMODULARMAX.

$c_u(A) = \sum_{i \in A} c_u(i)$, where $c_u(i)$ is the nonnegative feature $u$ of the centroid for cluster $i$, and can be interpreted as a nonnegative score for cluster $i$. We use TF-IDF features for 20newsgroup. For the other datasets, we train a corresponding neural networks on a small random subset of training data (e.g., hundreds of samples) for one epoch, and use the inputs to the last fully connected layer (whose outputs are processed by softmax to generate class probabilities) as features. Because we always use ReLU activations between layers, the features are all nonnegative and the submodularity of $F(A)$ follows as a consequence. These features are also used by mini-batch $k$-means to generate clusters for MCL and SPLD.

For MCL, we set the number of inner loop iterations to $p \leq 50$. For each dataset, we choose $p$ as the number among $\{10, 20, 50\}$ that reduces the training loss the most in the first few iterations of the outer loop, and then use that $p$ for the remaining iterations. As shown in Table 4, we use $p = 50$ for 20newsgroups, MNIST and Fashion-MNIST, and $p = 20$ for the other three datasets.

We consider five variants of MCL: 1) MCL($\Delta = 0, \lambda = 0, \gamma = 0$) having neither submodular regularization that promotes diversity nor scheduling of $k$ that increases hardness; 2) MCL($\Delta = 0, \lambda > 0, \gamma > 0$), which decreases diversity by exponentially reducing the weight $\lambda$ of the submodular regularization, but does not have any scheduling of $k$, i.e., $k$ is fixed during the algorithm; 3) MCL($\Delta > 0, \lambda > 0, \gamma = 0$), which only uses the scheduling of $k$ shown in Algorithm 1, but the diversity weight $\lambda$ is positive and fixed during the algorithm, i.e., with $\gamma = 0$; 4) MCL-RAND($r,q$), which randomly samples $r$ clusters as a training set $\hat{A}$ after every $q$ rounds of the outer loop in Algorithm 1, and thus combines both MCL and SGD; 5) MCL($\Delta > 0, \lambda > 0, \gamma > 0$), which uses the scheduling of both $\lambda$ and $k$ shown in Algorithm 1. We tried five different combinations of $\{q, r\}$ for MCL-RAND($r,q$) and five different $\Delta$ values for MCL($\Delta > 0, \lambda > 0, \gamma > 0$), and report the one with the smallest test error. Other parameters, such as the initial values for $\lambda$ and $k$, the values for $\gamma$ and $p$, and the total number of clusters are the same for different variants (the exact values of these quantities are given in Table 4 of the Appendix).

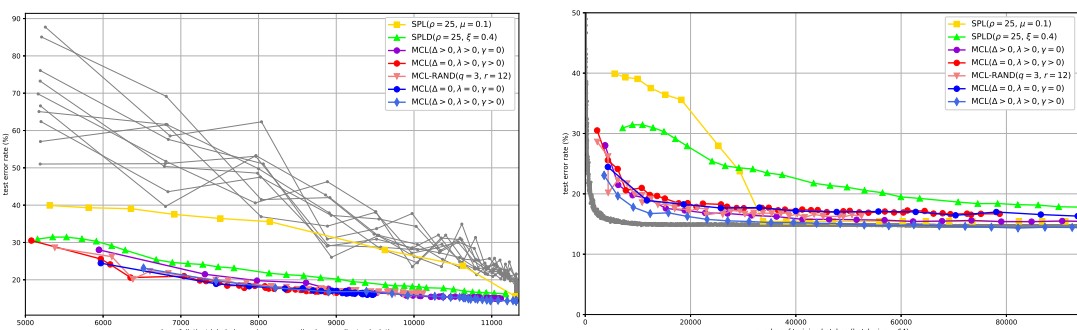

Figure 1: Test error rate (%) vs. number of distinct labeled samples ever needing loss gradient calculation (left) and number of training batches (right) on 20newsgroups (grey curves represents 10 random trials of SGD).

In MCL, running greedy is the only extra computation comparing to normal SGD. To show that in our implementation (see Section 4.6) its additional time cost is negligible, we report in Table 2 the total time cost for MCL($\Delta > 0, \lambda > 0, \gamma > 0$) and the time spent on our implementation WS-SUBMODULARMAX.

We summarize the main results in Figure 1-8. More results are given at the end of the appendix (Section 4.7). In all figures, grey curves correspond to the ten trials of SGD under a random curriculum. The legend in all figures gives the parameters used for the different methods using the following labels: 1) SPL ($\rho, \mu$); 2) SPLD($\rho, \xi$); and 3) MCL-RAND($q, r$).

Figures 1-6 show how the test error changes with (on the left) the number of distinct labeled samples ever needing a loss gradient calculation, and (on the right) the number of training batches,

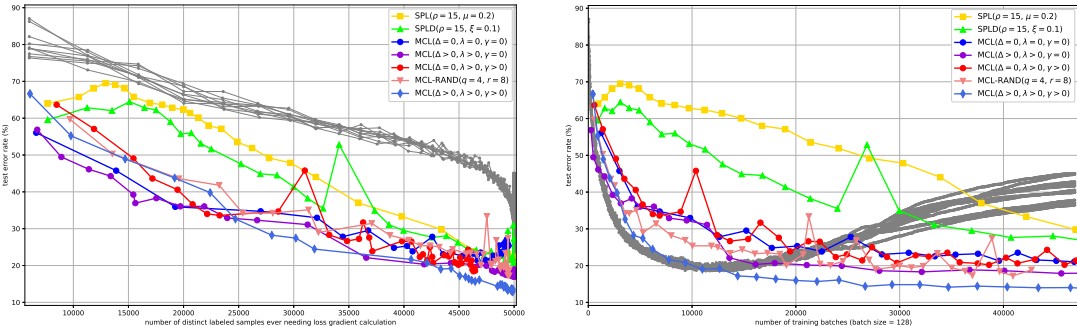

Figure 2: Test error rate (%) vs. number of distinct labeled samples ever needing loss gradient calculation (left) and number of training batches (right) on CIFAR10 (grey curves represents 10 random trials of SGD).

corresponding to training time. Note only MCL and its variants use the clustering trick, while SPL/SPLD need to compute loss on every sample and thus require knowledge of the labels of all samples. The left plot shows only the number of loss gradient calculations needed — 1) in MCL, for those clusters never selected in the curriculum, the loss (and hence the label) of only the centroid sample is needed; 2) in SPL/SPLD, for those samples never selected in the curriculum, their labels are needed only to compute the loss but not the gradient, so they are not reflected in the left plots of all figures because their labels are not used to compute a gradient. Therefore, thanks to the clustering trick, MCL and its variants can train without needing all labels, similar to semi-supervised learning methods. This can help to reduce the annotation costs, if an MCL process is done in tandem with a labeling procedure analogous to active learning. The right plots very roughly indicate convergence rate, namely how the test error decreases as a function of the amount of training.

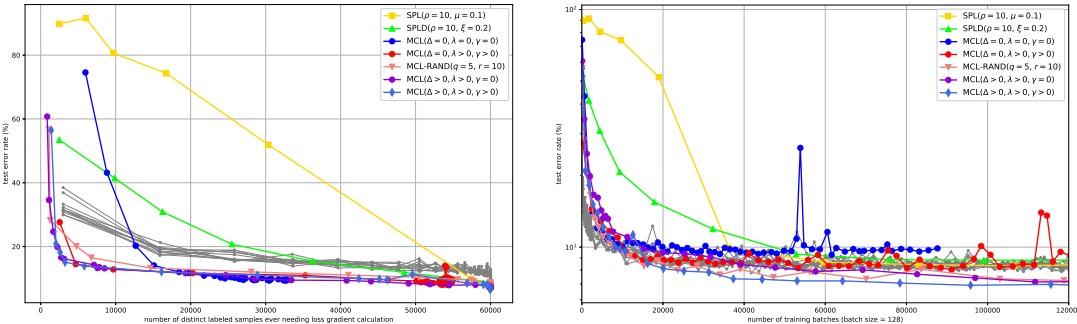

Figure 3: Test error rate (%) vs. number of distinct labeled samples ever needing loss gradient calculation (left) and number of training batches (right) on Fashion-MNIST (grey curves represents 10 random trials of SGD).

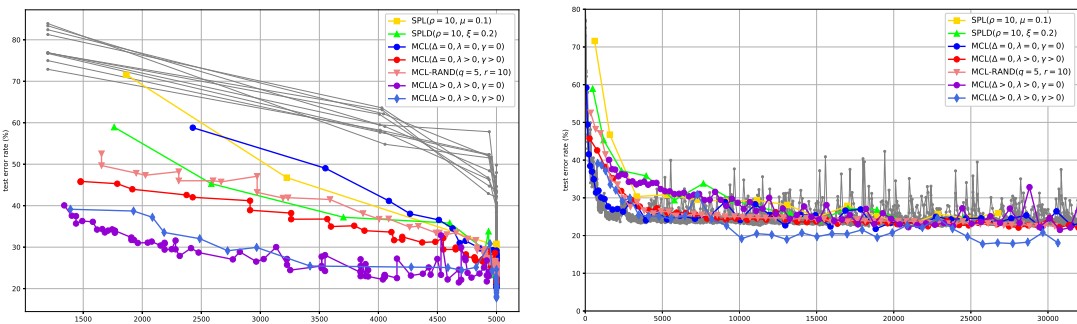

Figure 4: Test error rate (%) vs. number of distinct labeled samples ever needing loss gradient calculation (left) and number of training batches (right) on STL10 (grey curves represents 10 random trials of SGD).

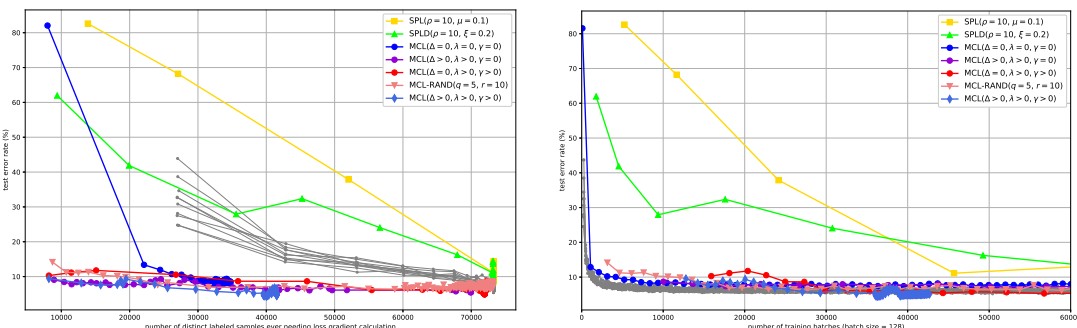

Figure 5: Test error rate (%) vs. number of distinct labeled samples ever needing loss gradient calculation (left) and number of training batches (right) on SVHN (grey curves represents 10 random trials of SGD).

On all datasets, MCL and most of its variants outperform SPL and SPLD in terms of final test accuracy (shown in Table 1) with comparable efficiency (shown in the right plots of all figures). MCL is slightly slower than SGD to converge in early stages but it can achieve a much smaller error when using the same number of labeled samples for loss gradients. Moreover, when using the same learning rate strategy, they can be more robust to overfitting, as shown in Figure 2. Comparing Figure 1 with Figure 2-6, MCL has the advantage when applied to deep models.

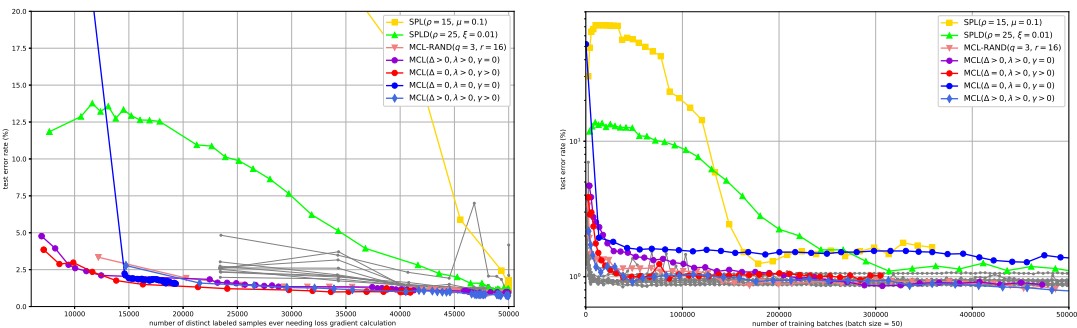

Figure 6: Test error rate (%) vs. number of distinct labeled samples ever needing loss gradient calculation (left) and number of training batches (right) on MNIST (grey curves represents 10 random trials of SGD).

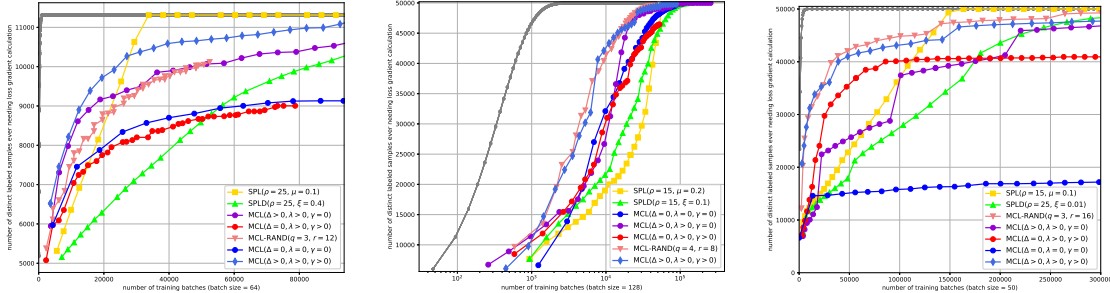

Figure 7: Number of distinct labeled samples ever needing loss gradient calculation vs. number of training batches for News20 (left), CIFAR10 (middle) and MNIST(right) (grey curves represents 10 random trials of SGD).

Among the five variants of MCL, MCL($\lambda > 0, \gamma > 0, \Delta > 0$) achieves the fastest convergence speed in later stages and the smallest final test error, while MCL($\Delta = 0, \lambda = 0, \Delta = 0$) usually achieves the worst performance (the only exception is on News20). Comparison between MCL($\Delta = 0, \lambda > 0, \gamma > 0$) and MCL($\lambda = 0, \gamma = 0, \Delta = 0$) shows that decreasing diversity improves the performance. MCL($\lambda > 0, \gamma > 0, \Delta > 0$) always outperforms MCL($\Delta = 0, \lambda > 0, \gamma > 0$). This indicates that increasing $k$ can bring advantages, e.g., more improvements in later stages.

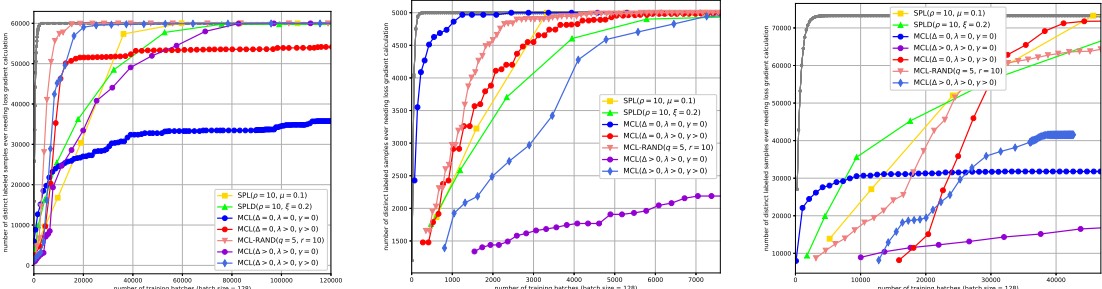

Figure 8: Number of distinct labeled samples ever needing loss gradient calculation vs. number of training batches for Fashion-MNIST (left), STL10 (middle) and SVHN (right) (grey curves represents 10 random trials of SGD).

MCL($\lambda > 0$, $\gamma > 0$, $\Delta > 0$) always outperforms MCL($\Delta > 0$, $\lambda > 0$, $\gamma = 0$), which supports our claim that it is better to decrease the diversity as training proceeds rather than keeping it fixed. In particular, MCL($\Delta > 0$, $\lambda > 0$, $\gamma = 0$) shows slower convergence than other MCL variants in later stages. In our experiments in the MCL($\Delta > 0$, $\lambda > 0$, $\gamma = 0$) case, we needed to carefully choose $\lambda$ and use a relatively large $\Delta$ for it to work at all, as otherwise it would repeatedly choose the same subset (with small $\Delta$, the loss term decreases as training proceeds, so with fixed $\lambda$ the diversity term comes to dominate the objective). This suggests that a large diversity encouragement is neither necessary nor beneficial when the model matures, possibly since $k$ is large at that point and there is ample opportunity for a diversity of samples to be selected just because $k$ is large, and also since encouraging too much loss-unspecific diversity at that point might only select outliers.

The combination of MCL and random curriculum (MCL-RAND) speeds up convergence, and sometimes (e.g., on MNIST, SVHN and Fashion-MNIST) leads to a good final test accuracy, but requires more labeled samples for gradient computation and still cannot outperform MCL($\lambda > 0$, $\gamma > 0$, $\Delta > 0$). These results indicate that the diversity introduced by submodular regularization does yield improvements, and changing both hardness and diversity improves performance.

Figure 7 and Figure 8 shows how the "number of distinct labeled samples ever needing loss gradient calculation" changes as training proceeds. It shows how the different methods trade-off between "training on more new samples" vs. "training on fewer distinct samples more often." Thanks to the clustering trick, MCL and its variants usually require fewer labeled samples for model training than SGD but more than SPL and SPLD.

**Acknowledgments** This work was done in part while author Bilmes was visiting the Simons Institute for the Theory of Computing in Berkeley, CA. This material is based upon work supported by the National Science Foundation under Grant No. IIS-1162606, the National Institutes of Health under award R01GM103544, and by a Google, a Microsoft, a Facebook, and an Intel research award. This work was supported in part by TerraSwarm, one of six centers of STARnet, a Semiconductor Research Corporation program sponsored by MARCO and DARPA.

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

# 4 APPENDIX

## 4.1 PROOF OF LEMMA 1

*Proof.* We have

$$\kappa_G = 1 - \min_{j \in V} \frac{L(j) + \lambda F(j|V \setminus j)}{L(j) + \lambda F(j)} = \lambda \cdot \max_{j \in V} \frac{F(j) - F(j|V \setminus j)}{L(j) + \lambda F(j)}$$

$$= \lambda \cdot \max_{j \in V} \frac{1 - \frac{F(j|V \setminus j)}{F(j)}}{\frac{L(j)}{F(j)} + \lambda} \leq \frac{\lambda \cdot \kappa_F}{\min_{j \in V} \frac{L(j)}{F(j)} + \lambda} = \frac{\kappa_F}{c_1/\lambda + 1}$$

Where $c_1 \triangleq \min_{j \in V} \frac{L(j)}{F(j)}$. $\qquad\square$

## 4.2 PROOF OF PROPOSITION 1

**Proposition 1.** *The maximum of multiple $\beta$-strongly convex functions is $\beta$-strongly convex as well.*

*Proof.* Let $g(x) = \max_i g_i(x)$, where $g_i(x)$ is $\beta$-strongly convex for any $i$. According to a definition of strongly convex function given in Theorem 2.1.9 (page 64) of (Nesterov, 2004), $\forall \lambda \in [0, 1]$, we have

$$g_i(\lambda x + (1 - \lambda)y) \leq \lambda g_i(x) + (1 - \lambda)g_i(y) - \frac{\beta}{2}\lambda(1 - \lambda)\|x - y\|_2^2, \forall i.$$

The following proves that $g(x)$ is also $\beta$-strongly convex:

$$g(\lambda x + (1 - \lambda)y) = \max_i g_i(\lambda x + (1 - \lambda)y)$$

$$\leq \max_i \left[\lambda g_i(x) + (1 - \lambda)g_i(y)\right] - \frac{\beta}{2}\lambda(1 - \lambda)\|x - y\|_2^2$$

$$\leq \max_i \lambda g_i(x) + \max_i(1 - \lambda)g_i(y) - \frac{\beta}{2}\lambda(1 - \lambda)\|x - y\|_2^2$$

$$= \lambda g(x) + (1 - \lambda)g(y) - \frac{\beta}{2}\lambda(1 - \lambda)\|x - y\|_2^2.$$

$\qquad\square$

## 4.3 PROOF OF THEOREM 1

*Proof.* The objective $g(w)$ of the minimax problem in Eq. (2) after eliminating $A$ is given in Eq. (4). Since $G(A)$ in Eq. (5) is monotone non-decreasing submodular, the optimal subset $A$ when defining $g(w)$ in Eq. (4) always has size $k$ if $|V| \geq k$. In addition, because the loss function $L(y_i, f(x_i, w))$ is $\beta$-strongly convex, $g(w)$ in Eq. (4) is the maximum over multiple $k\beta$-strongly convex functions with different $A$. According to Proposition 1, $g(w)$ is also $k\beta$-strongly convex, i.e.,

$$g(\hat{w}) \geq g(w^*) + \nabla g(w^*)^T(\hat{w} - w^*) + \frac{k\beta}{2}\|\hat{w} - w^*\|_2^2, \quad \forall \nabla g(w^*) \in \partial g(w^*). \qquad (14)$$

Since the convex function $g(w)$ achieves minimum on $w^*$, it is valid to substitute $\nabla g(w^*) = 0 \in \partial g(w^*)$ into Eq. (14). After rearrangement, we have

$$\|\hat{w} - w^*\|_2^2 \leq \frac{2}{k\beta}\left[g(\hat{w}) - g(w^*)\right]. \qquad (15)$$

In the following, we will prove $g(w^*) \geq \alpha \cdot g(\hat{w})$, which together with Eq. (15) will lead to the final bound showing how close $\hat{w}$ is to $w^*$.

Note $\hat{g}(w)$ (Eq. (6)) is a piecewise function, each piece of which is convex and associated with different $\hat{A}$ achieved by a submodular maximization algorithm of approximation factor $\alpha$. Since $\hat{A}$ is not guaranteed to be a global maxima, unlike $g(w)$, the whole $\hat{g}(w)$ cannot be written as the maximum of multiple convex functions and thus can be non-convex. Therefore, gradient descent in lines 6-9 of Algorithm 1 can lead to either: 1) $\hat{w}$ is a global minima of $\hat{g}(w)$; or 2) $\hat{w}$ is a local minima of $\hat{g}(w)$. Saddle points do not exist on $\hat{g}(w)$ because each piece of it is convex. We are also assuming other issues associated with the boundaries between convex pieces do not repeatedly occur.

1) When $\hat{w}$ is a global minima of $\hat{g}(w)$, we have
$$g(w^*) \geq \hat{g}(w^*) \geq \hat{g}(\hat{w}) \geq \alpha \cdot g(\hat{w}). \tag{16}$$
The first inequality is due to $g(\cdot) \geq \hat{g}(\cdot)$. The second inequality is due to the global optimality of $\hat{w}$. The third inequality is due to the approximation bound $\hat{g}(\cdot) \geq \alpha \cdot g(\cdot)$ guaranteed by the submodular maximization in Step 7 of Algorithm 1.

2) When $\hat{w}$ is a local minima of $\hat{g}(w)$, we have $\nabla \hat{g}(\hat{w}) = 0$. Let $h(w)$ be the piece of $\hat{g}(w)$ where $\hat{w}$ is located, then $\hat{w}$ has to be a global minima of $h(w)$ due to the convexity of $h(w)$. Let $\mathcal{A}$ denote the ground set of $\hat{A}$ on all pieces of $\hat{g}(w)$, we define an auxiliary convex function $\tilde{g}(w)$ as
$$\tilde{g}(w) \triangleq \max_{A \in \mathcal{A}} \sum_{i \in A} L\left(y_i, f(x_i, w)\right) + \lambda F(A). \tag{17}$$
It is convex because it is defined as the maximum of multiple convex function. So we have
$$\hat{g}(w) \leq \tilde{g}(w) \leq g(w), \forall w \in \mathbb{R}^m. \tag{18}$$
The first inequality is due to the definition of $\mathcal{A}$, and the second inequality is a result of $\mathcal{A} \subseteq 2^V$ by comparing $g(w)$ in Eq. (4) with $\tilde{g}(w)$ in Eq. (17). Let $\tilde{w}$ denote a global minima of $\tilde{g}(w)$, we have
$$g(w^*) \geq \tilde{g}(w^*) \geq \tilde{g}(\tilde{w}) \geq h(\tilde{w}) \geq h(\hat{w}) = \hat{g}(\hat{w}) \geq \alpha \cdot g(\hat{w}). \tag{19}$$
The first inequality is due to Eq. (18), the second inequality is due to the global optimality of $\tilde{w}$ on $\tilde{g}(w)$, the third inequality is due to the definition of $\tilde{g}(w)$ in Eq. (17) ($\tilde{g}(w)$ is the maximum of all pieces of $\hat{g}(w)$ and $h(w)$ is one piece of them), the fourth inequality is due to the global optimality of $\hat{w}$ on $h(w)$, the last inequality is due to the approximation bound $\hat{g}(\cdot) \geq \alpha \cdot g(\cdot)$ guaranteed by the submodular maximization in Step 7 of Algorithm 1.

Therefore, in both cases we have $g(w^*) \geq \alpha \cdot g(\hat{w})$. Applying it to Eq. (15) results in
$$\|\hat{w} - w^*\|_2^2 \leq \frac{2}{k\beta}\left(\frac{1}{\alpha} - 1\right) \cdot g(w^*). \tag{20}$$
$\square$

## 4.4 PROPOSITION 2

**Proposition 2.** *If $x \in [0, 1]$, the following inequality holds true.*
$$\frac{x}{1 - e^{-x}} - 1 \leq x. \tag{21}$$

*Proof.* Due to two inequalities $e^x \leq 1 + x + x^2/2$ for $x \leq 0$ and $1 - e^{-x} \geq x/2$ for $x \in [0, 1]$,
$$\frac{x}{1 - e^{-x}} - 1 = \frac{x - 1 + e^{-x}}{1 - e^{-x}} \leq \frac{x - 1 + (1 - x + x^2/2)}{x/2} = x. \tag{22}$$
$\square$

## 4.5 PROOF OF COROLLARY 1

*Proof.* Applying the inequality in Proposition 2 and the approximation factor of lazy greedy $\alpha = (1 - e^{-\kappa_G})/\kappa_G$ to the right hand side of Eq. (9) from Theorem 1 yields
$$\|\hat{w} - w^*\|_2^2 \leq \frac{2}{k\beta}\left(\frac{1}{\alpha} - 1\right) \cdot g(w^*)$$
$$= \frac{2}{k\beta}\left(\frac{\kappa_G}{1 - e^{-\kappa_G}} - 1\right) \cdot g(w^*) \leq \frac{2\kappa_G}{k\beta} \cdot g(w^*), \tag{23}$$
where $\kappa_G$ is the curvature of submodular function $G(\cdot)$ defined in Eq. (5). Substituting the inequality about $\kappa_G$ from Lemma 1 into Eq. (23) results in
$$\|\hat{w} - w^*\|_2^2 \leq \frac{2\kappa_F}{k\beta(c_1/\lambda + 1)} \leq \frac{2\kappa_F}{\beta c_1} \times \frac{\lambda}{k} \times g(w^*). \tag{24}$$
We use subscript as the index for iterations in the outer-loop, e.g., $\hat{w}_T$ is the model weights $w$ after the $T^{th}$ iteration of outer-loop. If we decrease $\lambda$ exponentially from $\lambda = \lambda_0$ and increase $k$ linearly from $k = k_0$, as Step 11 in Algorithm 1, we have
$$\|\hat{w}_T - w_T^*\|_2^2 \leq \frac{2\kappa_F \lambda_0}{\beta c_1} \times \frac{(1 - \gamma)^T}{(k_0 + T\Delta)} \times g(w_T^*), \tag{25}$$

According to the definition of $g(\cdot)$ in Eq. (4), for $g(w_T^*)$ we have

$$
\begin{aligned}
g(w_T^*) &= \min_{w \in \mathbb{R}^m} \max_{A \subseteq V, |A| \leq k} \sum_{i \in A} L\left(y_i, f(x_i, w)\right) + \lambda F(A) \\
&\leq \min_{w \in \mathbb{R}^m} \max_{A \subseteq V, |A| \leq k} \sum_{i \in A} L\left(y_i, f(x_i, w)\right) + \lambda_0 (1-\gamma)^T \max_{A \subseteq V, |A| \leq k} F(A) \\
&= g(w_\infty^*) + \lambda_0 (1-\gamma)^T c_2,
\end{aligned}
\tag{26}
$$

where

$$
g(w_\infty^*) \triangleq \min_{w \in \mathbb{R}^m} \max_{A \subseteq V, |A| \leq k} \sum_{i \in A} L\left(y_i, f(x_i, w)\right), c_2 \triangleq \max_{A \subseteq V, |A| \leq k} F(A).
\tag{27}
$$

Substituting Eq. (26) to Eq. (25) yields

$$
\|\hat{w}_T - w_T^*\|_2^2 \leq \frac{2\kappa_F \lambda_0}{\beta c_1} \times \frac{(1-\gamma)^T}{(k_0 + T\Delta)} \times \left[ g(w_\infty^*) + \lambda_0 c_2 (1-\gamma)^T \right],
\tag{28}
$$

If we can tolerate more expensive computational cost for running submodular maximization with larger budget $k$, and increase $k$ exponentially, i.e., $k \leftarrow (1 + \Delta) \cdot k$, we have

$$
\|\hat{w}_T - w_T^*\|_2^2 \leq \frac{2\kappa_F \lambda_0}{\beta c_1 k_0} \times \left( \frac{1-\gamma}{1+\Delta} \right)^T \times \left[ g(w_\infty^*) + \lambda_0 c_2 (1-\gamma)^T \right].
\tag{29}
$$

This completes the proof. $\qquad\square$

## 4.6 Submodular Maximization Starting from a Previous "Warm" Solution

Algorithm 1 repeatedly runs a greedy procedure to solve submodular maximization, and this occurs two nested loops deep. In this section we describe how we speed this process up.

Our first strategy reduces the size of the ground set before starting a more expensive submodular maximization procedure. We use a method described in (Wei et al., 2014a) where we sort the elements of $V$ non-increasingly by $G(i|V \setminus i)$ and then remove any element $i$ from $V$ having $G(i) < G(\delta(k)|V \setminus \delta(k))$ where $\delta(k)$ is $k^{\text{th}}$ element in the sorted permutation. Any such element will never be chosen by the $k$-cardinality constrained greedy procedure because for any $\ell \in \{1, 2, \ldots, k\}$, and any set $A$, we have $G(\delta(\ell)|A) \geq G(\delta(\ell)|V \setminus \delta(\ell)) \geq G(\delta(k)|V \setminus \delta(k)) > G(i) \geq G(i|A)$ and thus greedy would always be able to choose an element better than $i$. This method results in no reduction in approximation quality, although it might not yield any speedup at all. But with a decreasing $\lambda$, $G(A)$ becomes more modular, and the filtering method can become more effective. Other methods we can employ are those such as (Zhou et al., 2017; Mirzasoleiman et al., 2015), resulting in small reduction in approximation quality, but we do not describe these further.

The key contribution of this section is a method exploiting a potential warm start set that might already achieve a sufficient approximation quality. Normally, the greedy procedure starts with the empty set and adds elements greedily until a set of size $k$ is reached. In Algorithm 1, by contrast, a previous iteration has already solved a size-$k$ constrained submodular maximization problem for the previous submodular function, the solution to which is one that could very nearly already satisfy a desired approximation bound for the current submodular function. The reason for this is that, depending on the weight update method in line 9 of Algorithm 1 between inner loop iterations, and the changes to parameters $\Delta$ and $\gamma$ between outer iterations, the succession of submodular functions might not change very quickly. For example, when the learning rate $\eta$ is small, the $\hat{A}$ from the previous iteration could still be valued highly by the current iteration's function, so running a greedy procedure from scratch is unnecessary. Our method warm-starts a submodular maximization process with a previously computed set, and offers a bound that trades off speed and approximation quality.

The approach is given in Algorithm 2, which (after the aforementioned filtering in line 3 (Wei et al., 2014a)) tests in linear time if the warm start set $\hat{A}$ already achieves a sufficient approximation quality, and if so, possibly improves it further with an additional linear or quasilinear time computation. To test approximation quality of $\hat{A}$, our approach uses a simple modular function upper bound, in line 4, to compute an upper bound on the global maximum value. For the subsequent improvement of $\hat{A}$, our approach utilizes a submodular semigradient approach (Iyer et al., 2013) (specifically subgradients (Fujishige, 2005) in this case). If the warm-start set $\hat{A}$ does not achieve sufficient approximation quality in line 5, the algorithm backs off to standard submodular maximization in line

11 (we use the accelerated/lazy greedy procedure (Minoux, 1978) here although other methods, e.g., (Mirzasoleiman et al., 2015), can be used as well).

---

**Algorithm 2** Warm Start (WS) WS-SUBMODULARMAX($G, k, \hat{A}, \tilde{\alpha} \in [0, 1)$)

1: **Input:** $G(\cdot)$, $k$, $\hat{A}$, $\tilde{\alpha}$
2: **Output:** $\tilde{A}$
3: Reduce ground set size: arrange $V$ non-increasingly in terms of $G(i|V \backslash i)$ in a permutation $\delta$ where $\delta(k)$ is the $k^{th}$ element, set $V \leftarrow \{i \in V | G(i) \geq G(\delta(k)|V \backslash \delta(k))\}$;
4: Compute upper bound to maximum of Eq. (5):
$$\tau = \max_{A \in V, |A| \leq k} \sum_{i \in A} \left[ L\left(y_i, f(x_i, w^t)\right) + \lambda F(i) \right]$$
5: **if** $G(\hat{A}) \geq \tilde{\alpha} \cdot \tau$ **then**
6:     Permutation $\sigma$ of $V$: the first $k$ elements have $S_k^\sigma = \hat{A}$ and are chosen ordered non-increasing by $\kappa_G(v)$; the remaining $n - k$ elements $V \backslash \hat{A}$ for $\sigma$ are chosen non-increasing by $\kappa_G(v)$.
7:     Define modular function $h_{\hat{A}}(A) \triangleq \sum_{i \in A} h_{\hat{A}}(i)$ with $h_{\hat{A}}(\sigma(i)) = G(S_i^\sigma) - G(S_{i-1}^\sigma)$;
8:     Compute tight, at $\hat{A}$, lower bound $L(A)$ of $G(A)$:
$$L(A) \triangleq G(\hat{A}) + h_{\hat{A}}(A) - h_{\hat{A}}(\hat{A}) \leq G(A)$$
9:     $\tilde{A} \leftarrow \text{argmax}_{A \in V, |A| \leq k} L(A)$;
10: **else**
11:     $\tilde{A} \leftarrow \text{LAZYGREEDY}(G, V, k)$;
12: **end if**

---

Line 4 computes the upper bound $\tau \geq \max_{A \in V, |A| \leq k} G(A)$ which holds due to submodularity, requiring only a modular maximization problem (which can be done in $O(|V|)$ time, independent of $k$, to select the top $k$ elements). Line 5 checks if an $\tilde{\alpha}$ approximation to this upper bound is achieved by the warm-start set $\hat{A}$, and if not we back off to a standard submodular maximization procedure in line 11.

If $\hat{A}$ is an $\tilde{\alpha}$ approximation to the upper bound $\tau$, then lines 6-9 runs a subgradient optimization procedure, a process that can potentially improve it further. The approach selects a subgradient defined by a permutation $\sigma = (\sigma(1), \sigma(2), \ldots, \sigma(n))$ of the elements. The algorithm then defines a modular function $L(A)$, tight at $\hat{A}$ and a lower bound everywhere else, i.e., $L(\hat{A}) = G(\hat{A})$, and $\forall A, L(A) \leq G(A)$. Any permutation will achieve this as long as $\hat{A} = \{\sigma(1), \sigma(2), \ldots, \sigma(k)\}$. The specific permutation we use is described below. Once we have the modular lower bound, we can do simple and fast modular maximization.

Lines 6-9 of Algorithm 2 offer a heuristic that can only improve the objective — letting $\tilde{A}$ be the solution after line 9, we have
$$G(\tilde{A}) \geq L(\tilde{A}) \geq L(\hat{A}) = G(\hat{A}). \tag{30}$$
The first inequality follows since $L(\cdot)$ is a lower bound of $G(\cdot)$; the second inequality follows from the optimality of $\hat{A}^+$; the equality follows since $L$ is tight at $\hat{A}$.

The approximation factor $\tilde{\alpha}$ is distinct from the submodular maximization approximation factor $\alpha$ achieved by the greedy algorithm. Setting, for example $\tilde{\alpha} = 1 - 1/e$ would ask for the previous solution to be this good relative to $\tau$, the upper bound on the global maximum, and the algorithm would almost always immediately jump to line 11 since achieving such approximation quality might not even be possible in polynomial time (Feige, 1998). With $\tilde{\alpha}$ large, we recover the approximation factor of the greedy algorithm but ignore the warm start. If $\tilde{\alpha}$ is small, many iterations might use the warm start from the previous iteration, updating it only via one step of subgradient optimization, but with a worse approximation factor. In practice, therefore, we use a more lenient bound (often we set $\tilde{\alpha} = 1/2$) which is a good practical tradeoff between approximation accuracy and speed (meaning lines 6-9 execute a reasonable fraction of the time leading to a good speedup, i.e., in our experiments, the time cost for WS-SUBMODULARMAX increases if $\alpha = 1$ by a factor ranging from about 3 to 5). In general, we have the following final bound based on the smaller of $\tilde{\alpha}$ and $\alpha$.

**Lemma 2.** *Algorithm 2 outputs a solution $\hat{A}$ such that $G(\hat{A}) \geq \min\{\tilde{\alpha}, \alpha\} \times \max\limits_{A \in V, |A| \leq k} G(A)$, where $\alpha$ is the approximation factor of the greedy procedure (typically $\alpha = (1 - e^{-\kappa_G})/\kappa_G$).*

*Proof.* Let $A^*$ denote an optimal solution to Eq. (5):

$$A^* \in \operatorname*{argmax}_{A \in V, |A| \leq k} \sum_{i \in A} L\left(y_i, f(x_i, w^t)\right) + \lambda F(A). \tag{31}$$

$\tau$ computed in line 4 is an upper bound to $G(A^*)$ since:

$$\begin{aligned}
\tau &\geq \sum_{i \in A^*} \left[ L\left(y_i, f(x_i, w^t)\right) + \lambda F(i) \right] \\
&= \sum_{i \in A^*} L\left(y_i, f(x_i, w^t)\right) + \lambda \sum_{i \in A^*} F(i) \\
&\geq \sum_{i \in A^*} L\left(y_i, f(x_i, w^t)\right) + \lambda F(A^*).
\end{aligned} \tag{32}$$

The first inequality follows by the definition of $\tau$; the last inequality is due to submodularity, guaranteeing $F(i) \geq F(i|B)$ for any $B \subseteq V$.

When $G(\hat{A}) \geq \tilde{\alpha} \cdot \tau$ (line 5), the subgradient ascent can only improve the objective. Thus, we have $G(\hat{A}) \geq \tilde{\alpha} \cdot \max_{A \in V, |A| \leq k} G(A)$ for $\hat{A}$ obtained in line 9. Otherwise, we run the greedy algorithm on the reduced ground set $V$. Thus, we have $G(\hat{A}) \geq \alpha \cdot \max_{A \in V, |A| \leq k} G(A)$ for $\hat{A}$ obtained in line 11. $\qquad\square$

The heuristic in lines 6-9 is identical to one step of the semigradient-based minorization-maximization (MM) scheme used in, for example, (Narasimhan & Bilmes, 2005; Jegelka & Bilmes, 2011; Iyer & Bilmes, 2012; Iyer et al., 2013). Which permutation to use for the subgradient in order to tighten the gap has been an issue discussed as far back as (Narasimhan & Bilmes, 2005). In the present work, we offer a new heuristic for this problem. Let the first $i$ elements in the permutation $\sigma$ be denoted $S_i^\sigma = \{\sigma(1), \sigma(2), \ldots, \sigma(i)\}$, and let $A_{i-1}^\sigma \triangleq \{\sigma(j) \in A | j < i\} = S_{i-1}^\sigma \cap A \subseteq S_{i-1}^\sigma$ for any $i \in A$. The gap we wish to reduce is

$$0 \leq G(A) - L(A) = \sum_{\sigma(i) \in A} \left[ G(\sigma(i)|A_{i-1}^\sigma) - G(\sigma(i)|S_{i-1}^\sigma) \right] \tag{33}$$

$$= \sum_{\sigma(i) \in A} G(\sigma(i)) \cdot \left[ \frac{G(\sigma(i)|A_{i-1}^\sigma)}{G(\sigma(i))} - \frac{G(\sigma(i)|S_{i-1}^\sigma)}{G(\sigma(i))} \right] \tag{34}$$

$$\leq \sum_{\sigma(i) \in A} G(\sigma(i)) \cdot \left[ \frac{G(\sigma(i)|A_{i-1}^\sigma)}{G(\sigma(i))} - (1 - \kappa_G(\sigma(i))) \right] \tag{35}$$

Which follows since $h_{\hat{A}}(\sigma(i)) = G(\sigma(i)|S_{i-1}^\sigma)$ by definition. Line 6 chooses a particular permutation in an attempt to reduce this gap. Define an element-wise form of curvature as $\kappa_G(v) = 1 - G(v|V \backslash v)/G(v) \in [0, 1]$ for all $v \in V$. Note that $\kappa_G = \max_{v \in V} \kappa_G(v)$. If $\kappa_G(v) \approx 0$ then $G$ is practically modular at $v$ and so $G(v|A) \approx G(v)$ for any set $A$; in other words, $G(v|A)$ is close to $v$'s maximum possible gain even if $v$ is ranked with index very late in the permutation $\sigma$ where $A$ is very large. If $\kappa_G(v) \approx 1$, on the other hand, then there is some set $A \subseteq V \setminus \{v\}$ that can appreciably reduce $G(v|A)$ relative to the maximum possible gain $G(v)$, and so it is best to rank $v$ very early in the order $\sigma$ where $A$ must be a small set. One heuristic to achieve these goals is to choose a permutation $\sigma$ that arranges the elements in an order non-increasing according to $\kappa_G(v)$, meaning $\kappa_G(\sigma(1)) \geq \kappa_G(\sigma(2)) \geq \ldots$. Choosing this order is therefore an attempt to keep each of the conditional gains $G(\sigma(i)|S_{i-1}^\sigma)$ as close as possible to $\sigma(i)$'s maximum possible gain, $G(\sigma(i))$. This corresponds to an attempt to reduce Eq. (34) (and correspondingly close the $G(A) - L(A)$ gap) as much as possible. Line 6 of Algorithm 2 does this, subject to the requirement that the first $k$ elements of the permutation must correspond to $\hat{A}$ in order to be a subgradient. These tricks all help lines 6-9 produce a better updated approximate maximizer but at appreciably increased speed.

## 4.7  ADDITIONAL RESULTS

This section concludes by, in the form of tables and plots, providing more information about our experiments and experimental results for the algorithms mentioned above.

| Dataset | News20 | MNIST | CIFAR10 | STL10 | SVHN | Fashion |
|---|---|---|---|---|---|---|
| #Training | 11314 | 50000 | 50000 | 5000 | 73257 | 50000 |
| #Test | 7532 | 10000 | 10000 | 8000 | 26032 | 10000 |
| #Feature | 129791 | $28 \times 28$ | $32 \times 32 \times 3$ | $96 \times 96 \times 3$ | $32 \times 32$ | $28 \times 28$ |
| #Class | 20 | 10 | 10 | 10 | 10 | 10 |

Table 3: Details regarding the datasets.

| Dataset | News20 | MNIST | CIFAR10 | STL10 | SVHN | Fashion |
|---|---|---|---|---|---|---|
| $p$ | 50 | 50 | 20 | 20 | 20 | 50 |
| #cluster | 200 | 1000 | 1000 | 400 | 800 | 1000 |
| $\gamma$ | 0.05 | 0.05 | 0.05 | 0.2 | 0.2 | 0.05 |
| initial $k$ | 4 | 4 | 4 | 20 | 30 | 10 |
| initial $\lambda$ | $6 \times 10^{-6}$ | $1 \times 10^{-6}$ | $8 \times 10^{-7}$ | $1 \times 10^{-7}$ | $1 \times 10^{-6}$ | $1 \times 10^{-6}$ |
| initial $\eta$ | 3.5 | 0.02 | 0.01 | 0.02 | 0.01 | 0.02 |

Table 4: Parameters of MCL (Algorithm 1) and its variants for different datasets.

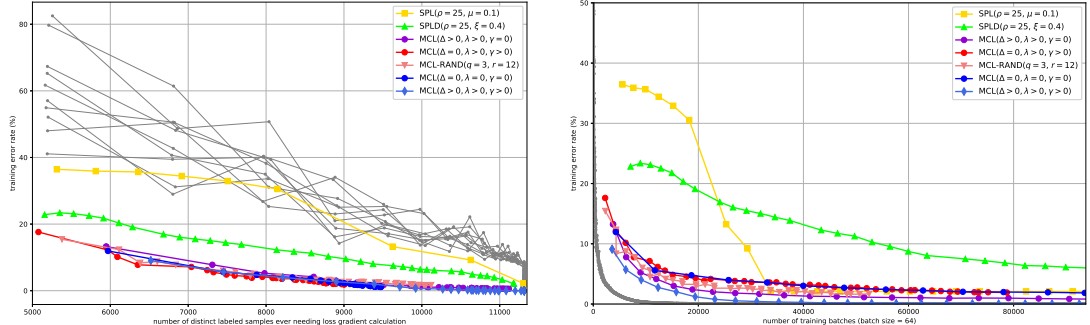

Figure 9: Training error rate (%) vs. number of distinct labeled samples ever needing loss gradient calculation (left) and number of training batches (right) on 20newsgroups (grey curves represents 10 random trials of SGD).

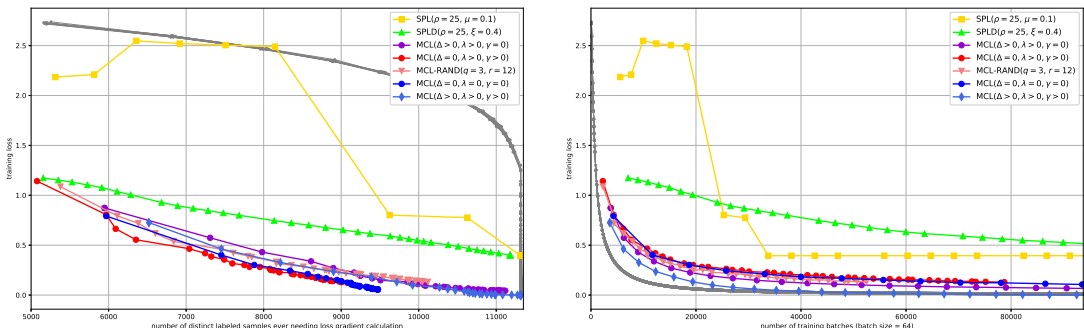

Figure 10: Training loss vs. number of distinct labeled samples ever needing loss gradient calculation (left) and number of training batches (right) on 20newsgroups (grey curves represents 10 random trials of SGD).

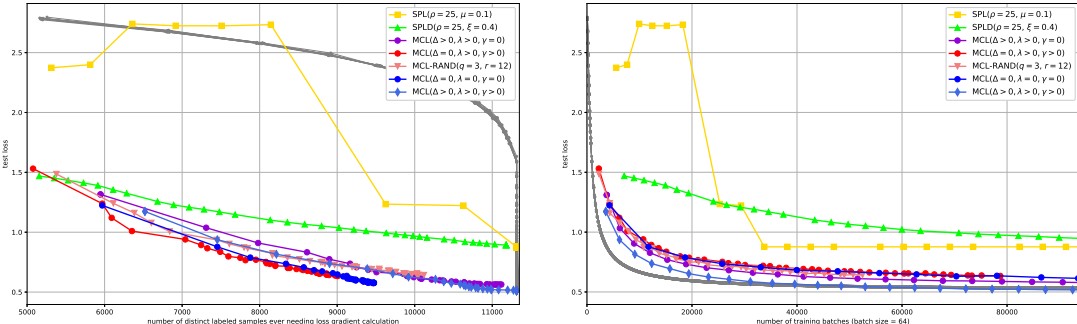

Figure 11: Test loss vs. number of distinct labeled samples ever needing loss gradient calculation (left) and number of training batches (right) on 20newsgroups (grey curves represents 10 random trials of SGD).

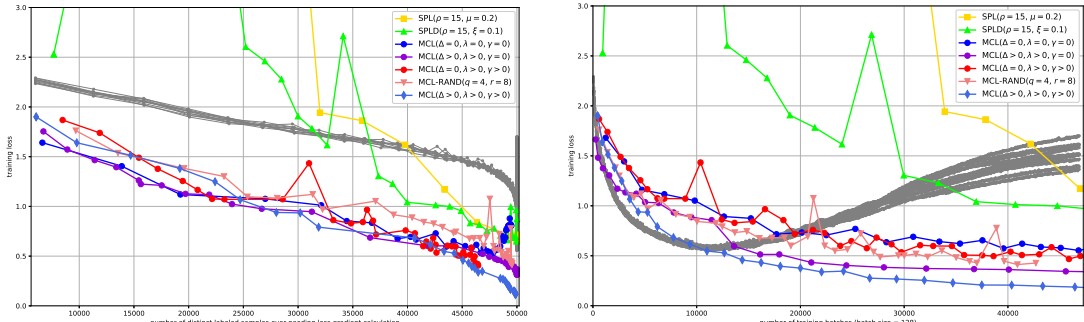

Figure 12: Training loss vs. number of distinct labeled samples ever needing loss gradient calculation (left) and number of training batches (right) on CIFAR10 (grey curves represents 10 random trials of SGD).

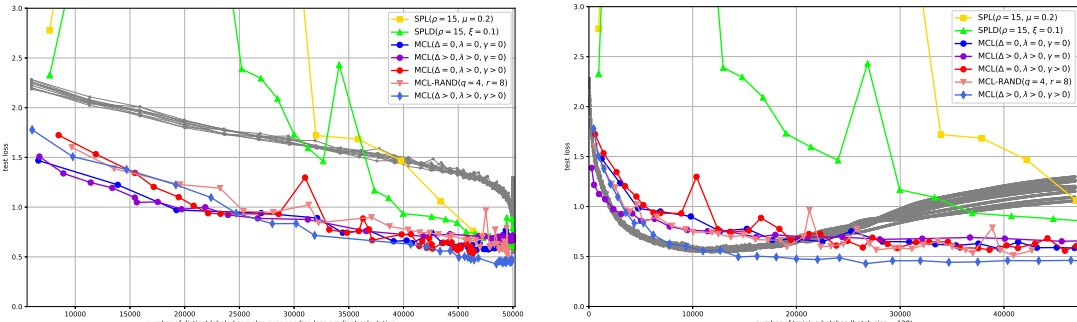

Figure 13: Test loss vs. number of distinct labeled samples ever needing loss gradient calculation (left) and number of training batches (right) on CIFAR10 (grey curves represents 10 random trials of SGD).

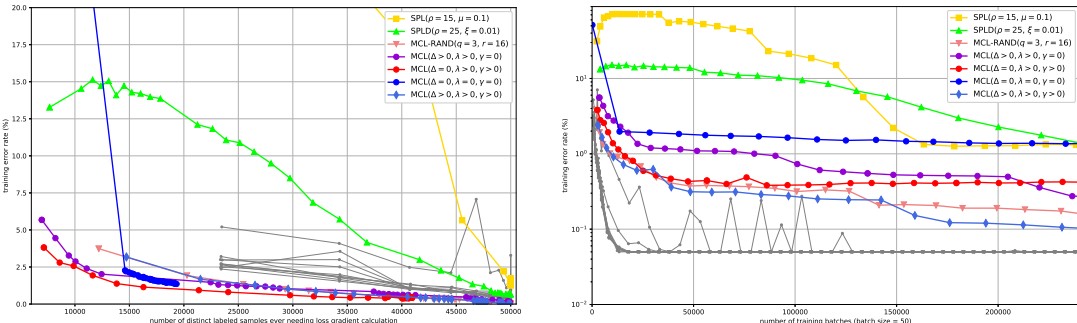

Figure 14: Training error rate (%) vs. number of distinct labeled samples ever needing loss gradient calculation (left) and number of training batches (right) on MNIST (grey curves represents 10 random trials of SGD).

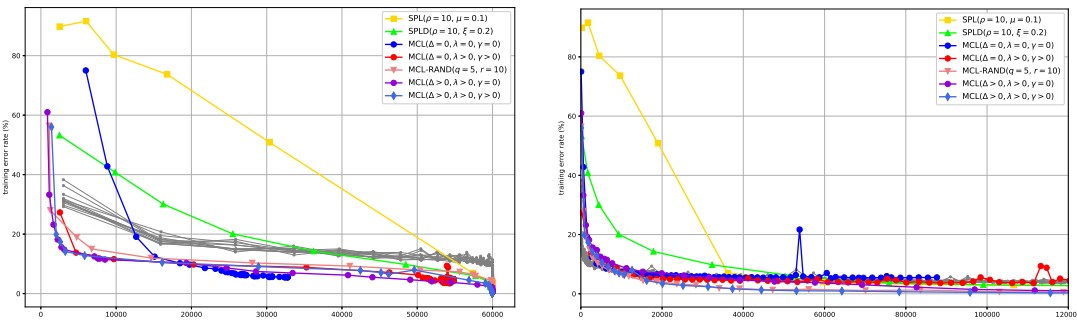

Figure 15: Training error rate (%) vs. number of distinct labeled samples ever needing loss gradient calculation (left) and number of training batches (right) on Fashion-MNIST (grey curves represents 10 random trials of SGD).

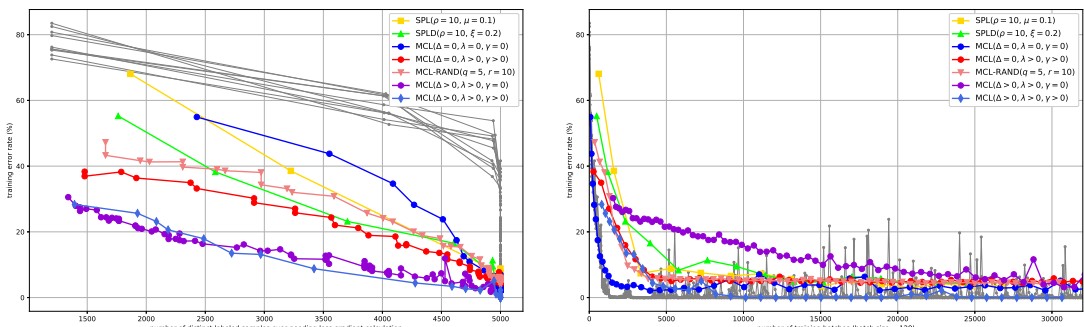

Figure 16: Training error rate (%) vs. number of distinct labeled samples ever needing loss gradient calculation (left) and number of training batches (right) on STL10 (grey curves represents 10 random trials of SGD).

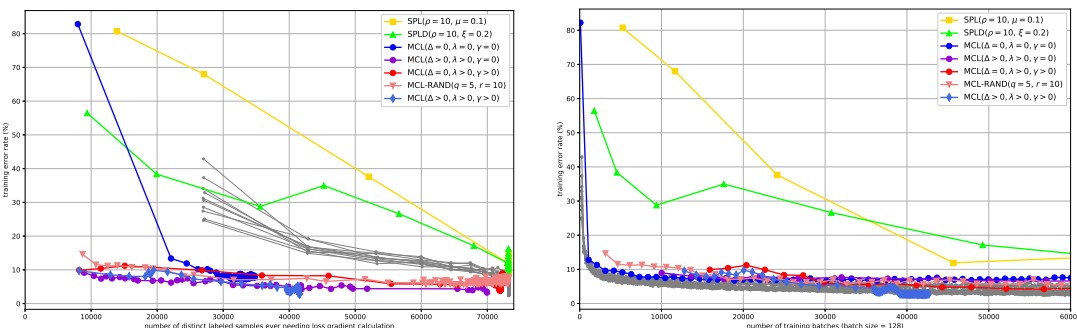

Figure 17: Training error rate (%) vs. number of distinct labeled samples ever needing loss gradient calculation (left) and number of training batches (right) on SVHN (grey curves represents 10 random trials of SGD).

