# OpenReview forum: "Minimax Curriculum Learning: Machine Teaching with Desirable Difficulties and Scheduled Diversity"
_ICLR.cc/2018/Conference — Accept (Poster)_

### Official Review · AnonReviewer2 · 2017-11-22
**Choosing diverse and hard training examples with submodular optimization.**

**Rating:** 5
**Confidence:** 3

**Review:**

Overview:
This paper proposes an approach to curriculum learning, where subsets of examples to train on are chosen during the training process. The proposed method is based on a submodular set function over the examples, which is intended to capture diversity of the included examples and is added to the training objective (eq. 2). The set is optimized to be as hard as possible (maximize loss), which results in a min-max problem. This is in turn optimized (approximately) by alternating between gradient-based loss minimization and submodular maximization. The theoretical analysis shows that if the loss is strongly convex, then the algorithm returns a solution which is close to the optimal solution. Empirical results are presented for several benchmarks.
The paper is mostly clear and the idea seems nice. On the downside, there are some limitations to the theoretical analysis and optimization scheme (see comments below).

Comments:
- The theoretical result (thm. 1) studies the case of full optimization, which is different than the proposed algorithm (running a fixed number of weight updates). It would be interesting to show results on sensitivity to the number of updates (p).
- The algorithm requires tuning of quite a few hyperparameters (sec. 3).
- Approximating a cluster with a single sample (sec. 2.3) seems rather crude. There should be some theoretical and/or empirical study of its effect on quality of the solution.

Minor/typos:
- what is G(j|G\j) in eq. (9)?
- why cite Anonymous (2018) instead of Appendix...?
- define V in Thm. 1.
- in eq. (4) it may be clearer to denote g_k(w). Likewise in eq. (6) \hat{g}_\hat{A}(w), and in eq. (14) \tilde{g}_{\cal{A}}(w).
- figures readability can be improved.

---

> ### Author Response · Authors · 2018-01-05
> **Newly added convergence bounds as functions of hyperparameters and fixed number of weight updates p, minors/typos revised**
>
> In the new revision, we've added a 4.5-page analysis to show the convergence speed of both outer-loop and the whole algorithm. A summary of the newly added analysis can be found in our new uploaded comments.
>
> Reply to Comments:
>
> Theorem 3 analyzes the convergence rate of the whole algorithm presented in Algorithm 1 with a fixed number of weight updates $p$ in each inner-loop. The first term in the bound exponentially decreases with power $p$.
>
> The convergence bounds in both Theorem 2 and Theorem 3 are functions of all hyperparameters $\lambda$, $\Delta$ and $p$. They show that exponentially decreasing $\lambda$ is sufficient to guarantee a linear rate of convergence, while choosing small $\Delta$ and $p$ make the algorithm efficient in computation. These theoretical analysis allows us to tune the hyperparameters in relatively small ranges.
>
> Instead of representing the whole cluster by the centroid everywhere, we only represent the hardness of a cluster by the loss on its centroid. By setting the number of clusters to be a large value, e.g., 1000 clusters for 50000 samples in our experiments, this hardness representation is accurate enough. It not only saves computation spent on submodular maximization in practice, but also makes the algorithm more robust to outliers, because it avoids selecting a single (or a few number of) outliers with extremely large loss.
>
> Reply to Minor/typos:
>
> G(j|G\j) contains a typo, it should be G(j|V\j)=G(V)-G(V\j), the marginal gain of element j conditioned on all the other elements in ground set V except j. Thanks for pointing this out!
>
> In the revision, 1) we changed all citations to Anonymous (2018) to specific sections in Appendix; 2) we define V in Theorem 1 and all other Theorems; 3) for simplicity of representation, we use g() without subscript when it causes no confusion. For example, Theorem 1 and Lemma 2 holds for any iteration in outer-loop, so we ignore the subscript of g(). When discussing relationship between different iterations of outer-loop, we add subscript to w in g(w) (e.g., in proof of Theorem 2) or add subscipt to g() (e.g., in proof of Theorem 3).

---

### Official Review · AnonReviewer3 · 2017-12-01
**Good theoretical results, but would have liked a stronger empirical story**

**Rating:** 6
**Confidence:** 4

**Review:**

This paper introduces MiniMax Curriculum learning, as an approach for adaptively train models by providing it different subsets of data. The authors formulate the learning problem as a minimax problem which tries to choose diverse example and "hard" examples, where the diversity is captured via a Submodular Loss function and the hardness is captured via the Loss function. The authors formulate the problem as an iterative technique which involves solving a minimax objective at every iteration. The authors argue the convergence results on the minimax objective subproblem, but do not seem to give results on the general problem. The ideas for this paper are built on existing work in Curriculum learning, which attempts to provide the learner easy examples followed by harder examples later on. The belief is that this learning style mimics human learners.

Pros:
- The analysis of the minimax objective is novel and the proof technique introduces several interesting ideas.
- This is a very interesting application of joint convex and submodular optimization, and uses properties of both to show the final convergence results
- Even through the submodular objective is only approximately solvable, it still translates into a convergence result
- The experimental results seem to be complete for the most part. They argue how the submodular optimization does not really affect the performance and diversity seems to empirically bring improvement on the datasets tried.

Cons:
- The main algorithm MCL is only a hueristic. Though the MiniMax subproblem can converge, the authors use this in somewhat of a hueristic manner.
- It seems somewhat hand wavy in the way the authors describe the hyper parameters of MCL, and it seems unclear when the algorithm converge and how to increase/decrease it over iterations
- The objective function also seems somewhat non-intuitive. Though the experimental results seem to indicate that the idea works, I think the paper does not motivate the loss function and the algorithm well.
- It seems to me the authors have experimented with smaller datasets (CIFAR, MNIST, 20NewsGroups). This being mainly an empirical paper, I would have expected results on a few larger datasets (e.g. ImageNet, CelebFaces etc.), particularly to see if the idea also scales to these more real world larger datasets.

Overall, I would like to see if the paper could have been stronger empirically. Nevertheless, I do think there are some interesting ideas theoretically and algorithmically. For this reason, I vote for a borderline accept.

---

> ### Author Response · Authors · 2018-01-05
> **Newly added theoretical analysis provides complete analysis of the whole algorithm, instructions on hyperparameter tuning, and supports the intuition behind the objective function**
>
> In the new revision, we add 4.5-page analysis to show the convergence speed for both the outer-loop and the whole algorithm. A summary of the newly added analysis can be found in our new uploaded comments.
>
> Reply to Cons:
>
> Theorem 3 in the new revision gives the convergence analysis for the whole algorithm, each of whose inner-loop uses fixed number of updates to approximately solve a minimax problem. It does not only show convergence, but also shows convergence rate for both the inner-loop and outer-loop.
>
> In Theorem 2 and Theorem 3, we show convergence bounds as functions of all hyperparameters. These results give strong intuition for how to choose the hyperparameters. They show that exponentially decreasing $\lambda$ is sufficient to guarantee a linear rate of convergence, while choosing small $\Delta$ and $p$ make the algorithm efficient computationally. In practice, we use grid search with small ranges to achieve the hyperparameters used in experiments.
>
> The intuitions behind the objetive function can be found in the two paragraphs above Section 1.1, the last two paragraphs of Section 1.1, and the first paragraph of Section 2. In these places, we provide evidence based on the nature of machine learning model/algorithms, the similarity to the human teaching/learning process, and the comparison to previous works. In addition, the objective function has nice theoretical properties. Our newly added theoretical analysis supports that decreasing diversity weight $\lambda$ and increasing hardness $k$ can improve the convergence bound. This provides further theoretical support.
>
> Our experiments verify several advantages of the proposed minimax curriculum learning across three different models and datasets. Our basic goal is to prove the idea of decreasing diversity and increasing hardness for general machine learning problems. This idea has never been studied before, either theoretically or empirically, as far as we know. We are working on experiments for much larger datasets such as ImageNet and COCO, and will make the results available as soon as we can.

---

### Official Review · AnonReviewer1 · 2017-12-05
**Good theoretical result on combining submodular set optimization with curriculum learning**

**Rating:** 6
**Confidence:** 3

**Review:**

The main strength of this paper, I think, is the theoretical result in Theorem 1. This result is quite nice. I wish the authors actually concluded with the following minor improvement to the proof that actually strengthens the result further.

The authors ended the discussion on thm 1 on page 7 (just above Sec 2.3) by saying what is sufficiently close to w*. If one goes back to (10), it is easy to see that what converges to w* when one of three things happen (assuming beta is fixed once loss L is selected).

1) k goes to infinity
2) alpha goes to 1
3) g(w*) goes to 0

The authors discussed how alpha is close to 1 by virtue of submodular optimization lower bounds there for what is close to w*. In fact this proof shows the situation is much better than that.

If we are really concerned about making what converge to w*, and if we are willing to tolerate the increasing computational complexity associated solving submodular problems with larger k, we can schedule k to increase over time which guarantees that both alpha goes to 1 and g(w*) goes to zero.

There is also a remark that G(A) tends to be modular when lambda is small which is useful.
From the algorithm, it seems clear that the authors recognized these two useful aspects of the objective and scheduled lambda to decrease exponentially and k to increase linearly.

It would be really nice to complete the analysis of Thm1 with a formal analysis of convergence speed for ||what-w*|| as lambda and k are scheduled in this fashion. Such an analysis would help practitioners make better choices for the hyper parameters gamma and Delta.

---

> ### Author Response · Authors · 2018-01-05
> **Newly added 4.5-page theoretical to the convergence rate of the whole algorithm, extend Theorem 1 to show outer-loop convergence rate**
>
> Thanks for your positive comments about the theoretical analysis and helpful suggestions to extend Theorem 1! In the new revision, we've added a 4.5-page analysis. This does not only complete the analysis of Theorem 1, but also shows the convergence speed for both the outer-loop and the whole algorithm, and show bounds as functions of hyperparameters. The results support our scheduling strategy for $\lambda$ and $k$. A summary of the newly added analysis can be found in our new uploaded comments above your review comments.

---

### Author Response · Authors · 2018-01-05
**Summary of newly added 4.5-page complete theoretical analysis to the convergence rate of the whole algorithm and hyperparameters**

We note that both Reviewer2 and Reviewer3 wish to see an analysis of the whole algorithm, and more details on hyperparameter tuning issues. Reviewer1 also provides helpful suggestions on how to strengthen Theorem 1's result. In fact, more complete theoretical analysis is the main concern of all reviewers. In the new revision, we've added a 4.5-page mathematical analysis giving a convergence rate of the whole algorithm with the scheduling of $k$ and $\lambda$. The result also shows how to set hyperparameters to change the convergence. Here is a summary.

1) Theorem 2 shows that either decreasing $\lambda$ exponentially or increasing $k$ exponentially results in a linear convergence rate for the outer-loop of our algorithm. It also shows that using a scheduling with decreasing $\lambda$ or/and increasing $k$ can gradually improve the bound. This supports our intuition of decreasing diversity and increasing hardness.

2) Theorem 3 gives the convergence rate of the whole algorithm (each inner-loop runs only $p$ iterations). It shows linear convergence rate for both the inner-loop and outer-loop. The bound has two terms, one decreases exponentially with power $p$ (#iterations for inner-loop) and the other decreases exponentially with power $T$ (#iterations for outer-loop).

3) Convergence bounds in both Theorem 2 and Theorem 3 contains all the hyperparameters $\gamma$, $\Delta$ and $p$. They show how the bounds change with these hyperparameters, and can help to choose hyperparameters in practice. For example, they show that exponentially decreasing $\lambda$ is sufficient to guarantee a linear rate of convergence, while choosing small $\Delta$ (the additive $k$ increment) and $p$ make the algorithm efficient in computation.

4) Potentially interesting to future analysis of more general continuous-combinatorial optimization: The constant factors in Theorem 2 implies that $\kappa_F/\beta$ (ratio between the curvature of submodular term and the strongly-convex constant of loss term) and $c_1$ (the minimal ratio between loss and singular gain over all samples) are two important quantities in analyzing convex-submodular hybrid optimization. The constant factor $c$ in Theorem 3 is a weighted sum of the optimal objective value of the minimax problem without the submodular term, and the maximal value for the submodular term only. It relates the convergence bound to the solutions of the two extreme cases of Eq.(2).

---

### Decision · Program_Chairs · 2018-01-29
**ICLR 2018 Conference Acceptance Decision**

**Decision:**

Accept (Poster)

**Comment:**

The submission formulates self paced learning as a specific iterative mini-max optimization, which incorporates both a risk minimization step and a submodular maximization for selecting the next training examples.

The strengths of the paper lie primarily in the theoretical analysis, while the experiments are somewhat limited to simple datasets: News20, MNIST, & CIFAR10.  Additionally, the main paper is probably too long in its current form, and could benefit from some of the proof details being moved to the appendix.